# Interaction of a Homologous Series of Amphiphiles with P-glycoprotein in a Membrane Environment—Contributions of Polar and Non-Polar Interactions

**DOI:** 10.3390/pharmaceutics15010174

**Published:** 2023-01-03

**Authors:** Maria João Moreno, Hugo A. L. Filipe, Susana V. P. Cunha, Cristiana V. Ramos, Patrícia A. T. Martins, Biebele Abel, Luís M. S. Loura, Suresh V. Ambudkar

**Affiliations:** 1Coimbra Chemistry Center-Institute of Molecular Sciences (CQC-IMS), Department of Chemistry, University of Coimbra, 3004-535 Coimbra, Portugal; 2CNC—Center for Neuroscience and Cell Biology, University of Coimbra, 3004-535 Coimbra, Portugal; 3CPIRN-IPG—Center of Potential and Innovation of Natural Resources, Polytechnic of Guarda, 6300-559 Guarda, Portugal; 4Laboratory of Cell Biology, CCR, National Cancer Institute, NIH, Bethesda, MD 20892, USA; 5Henry M. Jackson Foundation for the Advancement of Military Medicine, Inc., Uniformed Services University of the Health Science, Bethesda, MD 20817, USA; 6Faculty of Pharmacy, University of Coimbra, 3000-548 Coimbra, Portugal

**Keywords:** efflux transporters, MDR1, ABCB1, pharmacokinetics, MD simulations, partition coefficient, binding affinity, amphiphilic moment

## Abstract

The transport of drugs by efflux transporters in biomembranes limits their bioavailability and is a major determinant of drug resistance development by cancer cells and pathogens. A large number of chemically dissimilar drugs are transported, and despite extensive studies, the molecular determinants of substrate specificity are still not well understood. In this work, we explore the role of polar and non-polar interactions on the interaction of a homologous series of fluorescent amphiphiles with the efflux transporter P-glycoprotein. The interaction of the amphiphiles with P-glycoprotein is evaluated through effects on ATPase activity, efficiency in inhibition of [^125^I]-IAAP binding, and partition to the whole native membranes containing the transporter. The results were complemented with partition to model membranes with a representative lipid composition, and details on the interactions established were obtained from MD simulations. We show that when the total concentration of amphiphile is considered, the binding parameters obtained are apparent and do not reflect the affinity for P–gp. A new formalism is proposed that includes sequestration of the amphiphiles in the lipid bilayer and the possible binding of several molecules in P–gp’s substrate-binding pocket. The intrinsic binding affinity thus obtained is essentially independent of amphiphile hydrophobicity, highlighting the importance of polar interactions. An increase in the lipophilicity and amphiphilicity led to a more efficient association with the lipid bilayer, which maintains the non-polar groups of the amphiphiles in the bilayer, while the polar groups interact with P–gp’s binding pocket. The presence of several amphiphiles in this orientation is proposed as a mechanism for inhibition of P-pg function.

## 1. Introduction

The efflux protein P-glycoprotein (P–gp) is a membrane protein that actively transports small molecules from inside to the outside of cells, thus decreasing their effective concentration in the cells. This efflux protein is highly abundant in the apical membrane of endothelial cells from tight endothelia [1,2,3,4], imposing further difficulties in the permeation through those biological barriers. It is also over-expressed in drug-resistant cancer cells, being one of the responsible factors for the observed resistance to chemotherapy [5,6,7]. The currently accepted mechanism of transport by P–gp considers binding of the substrate from the cell membrane (with the protein in the open, inner facing conformation), followed by a conformational change that exposes the binding site to the outer leaflet of the membrane, from which the substrate may equilibrate with the outer aqueous media or with the outer leaflet of the membrane [8,9,10]. This conformational transition is coupled with the hydrolysis of ATP in a complex mechanism that, although widely studied, is not completely elucidated [11,12,13,14,15,16,17,18,19].

In spite of extensive work over the last three decades, the basis for the broad substrate specificity of P–gp is not well known, the latter interacting with molecules with a wide range of properties. A striking observation is that P–gp inhibitors are usually more hydrophobic than P–gp substrates [20,21,22,23,24]. The broad specificity of P–gp is usually considered a result from its large and flexible binding pocket, able to interact with several molecules possessing similar or distinct properties, with a competitive or non-competitive mechanism [25,26,27,28,29,30,31]. An intriguing aspect of P–gp (and other efflux proteins) is the activation or inhibition of their activity by the same molecule depending on its concentration [27,28,32], with the transition between activation to inhibition depending strongly on the modulator hydrophobicity [32]. The medium where P–gp is embedded (native membranes, micelles, or liposomes) also influences the behavior of the protein and how it interacts with other molecules,. [33,34,35,36,37] with some molecules switching between substrates and inhibitors depending on P-gp’s environment [38]. 

In this work, we address P–gp specificity and activity modulation with the use of a homologous series of fluorescent amphiphiles (NBD–Cn) that has a common polar moiety (the fluorescent NBD group) and an alkyl chain of variable length. This is complemented with the lysophospholipid NBD–LysoMPE, with 14 carbons in its acyl chain and the fluorescent NBD group in the polar phosphoethanolamine polar head group. The interaction with P–gp in native membranes from Hi-Five insect cells is accessed through studies of P–gp ATPase activity, as well as competition with the prazosin analog [^125^I]-IAAP for binding to P–gp. The association of the amphiphiles with the lipid portion of the membranes and with the whole native membranes is directly assessed, leading to their relative affinities for the lipid bilayer and membrane proteins. Molecular dynamics (MD) simulations have also been performed, providing important insights regarding the location of the NBD amphiphiles in the membrane, and when associated with P–gp, as well as on their relative affinity for the lipid bilayer and the protein.

The goal is to understand the role of polar and non-polar interactions in the binding affinity to P–gp and whether binding leads to P–gp activation or inhibition. It is shown that, when binding to P–gp is characterized using the commonly used formalisms (considering the total concentration of ligand), the binding parameters obtained are apparent and do not reflect the affinity of the ligand to P–gp. In contrast, in this work, the local concentration of the ligand in the distinct media (aqueous, lipid bilayer, and P–gp) is considered. The local concentrations were calculated from partition coefficients obtained directly from the aqueous media towards the native membranes and to model membranes with the representative lipid composition, which has been recently characterized by us [39]. This local concentration is then used to quantitatively characterize the intrinsic binding affinity of the NBD amphiphiles towards P–gp in the membrane environment. Another important goal of this work is to evaluate how the binding of several modulator molecules in P–gp’s binding pocket affects the transporter properties. For this goal we use a formalism recently proposed by us [40] that allows the explicit consideration of several binding sites on the protein, as well as the sequestration of the modulator in the lipid portion of the membrane. 

## 2. Materials and Methods

### 2.1. Materials

The lipids 1-palmitoyl-2-oleoyl-*sn*-glycero-3-phosphocholine (POPC), 1-palmitoyl-2-oleoyl-*sn*-glycero-3-phosphoethanolamine (POPE), and 1-palmitoyl-2-oleoyl-*sn*-glycero-3-phospho-L-serine (POPS) were acquired from Avanti Polar Lipids, Inc. (Alabaster, AL, USA). [^125^I] Iodoarylazidoprazosin (IAAP) 1100 Ci/mmol) was purchased from PerkinElmer Life Sciences (Wellesley, MA, USA), and Tariquidar was purchased from AdooQ BioScience (Irvine, CA, USA). The fluorescent NBD amphiphiles were synthesized and purified, as previously reported by us [41,42], and the products were stored at −20 °C in the dry state or dissolved in DMSO. The concentration of the NBD–amphiphiles was calculated from their absorption at the maxima (≅466 nm, ε = 2.1 × 10^4^) [41,42]. Reagents and solvents used were analytical grade or with higher purity, and water was distilled and further deionized with a final resistance ≥ 18 MΩ.

The structures and common molecular descriptors of the P–gp modulators characterized experimentally in this work are shown in Table 1.

### 2.2. Total Membrane Vesicle Preparation

High-Five insect cells (Invitrogen) were infected with recombinant baculovirus carrying mouse P–gp (mP–gp) gene (Abcb1a). Membrane vesicles were prepared by hypotonic lysis of the insect cells, followed by ultracentrifugation to collect the membrane vesicles, as described in detail in reference [43]. The native membranes obtained were suspended in storage buffer (45 mM Tris-HCl pH 7.5, 45 mM D-mannitol, 1.8 mM EGTA, 0.45% *v/v* aprotinin, 1.8 mM DTT, 0.9 mM AEBSF, and 10% *v/v* glycerol), aliquoted, and stored at −80 °C or in liquid nitrogen until use. The concentration of protein in the native membranes was quantified by the Sherman and Weismann method using Amido Black B dye [44].

### 2.3. ATPase Assays

The ATPase activity of P–gp in the High-Five insect cell membranes was measured by the end point P_i_ release assays, as described earlier with minor modifications [38]. Native membranes were diluted in ATPase assay buffer (final concentrations: 41.2 mM Tris-MES pH = 6.8, 50 mM KCl, 5 mM NaN_3_, 2 mM EGTA, 1.03 mM Ouabain, 2 mM DTT, and 10 mM MgCl_2_), and the modulators were added from stocks in DMSO (final DMSO 1% *v/v*). The final concentration of protein was quantified, being ca. 0.1 mg/mL. Four samples were prepared at each modulator concentration, containing ATP at 5 mM, two of which also contained vanadate at 0.3 mM. P–gp specific activity was measured as vanadate-sensitive ATPase activity. The basal rate of ATP hydrolysis was 67 ± 4 nmol Pi mgP^−1^ min^−1^, with 37 ± 4 nmol Pi mgP^−1^ min^−1^ being sensitive to vanadate. The results obtained for the effect of the modulators on the ATPase activity in the presence and absence of vanadate are shown in the Appendix A. No significant variation is observed in the presence of vanadate, indicating that the modulators’ effect is mediated by P–gp.

The results were first analyzed with the usual equation considering the total concentration of modulator and both activation and inhibition of ATPase activity by the modulators [45,46], see Appendix A. The results were then analyzed with the formalism derived in this manuscript considering the distribution of the modulator in the distinct environments (aqueous phase, lipid bilayer and P-gp) and several equal and independent binding sites in P-gp’s binding pocket, Equations (3)–(9). The best fit was obtained with nonlinear least-square analysis using Microsoft Excel [47].

### 2.4. Photolabeling of P–gp with [^125^I]-IAAP

The native membranes of High-Five insect cells expressing mP–gp (1.0 mg/mL) were incubated at room temperature with the indicated concentrations of the inhibitor for 5 min. IAAP (5 to 7 nM) was later added, and the reaction mixture was further incubated for 5 min under subdued light. The samples were photocross-linked with 366 nm UV light for 10 min at room temperature, followed by electrophoresis and quantification, as described previously [48]. The raw data are shown in the Appendix A. The results were first analyzed with the usual equation, considering the total concentration of IAAP and NBD amphiphile, as well as competitive inhibition of IAAP binding, see Appendix A. The dissociation constant considered for IAAP was taken from the literature, 412 nM [15]. The results were also analyzed with a formalism derived in this manuscript considering the distribution of the NBD amphiphile in the distinct environments (aqueous phase, lipid bilayer, and P–gp), the distribution of IAAP between the aqueous media and P–gp, and several equal and independent binding sites in P–gp’s binding pocket, Equation (10).

### 2.5. Preparation of Large Unilamelar Vesicles

The model membranes with a phospholipid composition representative of that of the native membranes (POPE:POPC:POPS 45:35 20) [39] were prepared from stock solutions of the required lipids in chloroform or in chloroform/methanol (87/13, *v/v*), as described before [49]. The solvent-free lipid residue was hydrated at 40 °C with previously heated simplified ATPase assay buffer (with all components except Ouabain and DTT). Large Unilamelar Vesicles were prepared by extrusion through two-staked polycarbonate filters (100 nm pore diameter), and the size of the LUVs was analyzed by dynamic light scattering (Zetasizer Nano ZS, Malvern, UK) being close to the filter pore size.

### 2.6. Association of the NBD Amphiphiles with Model Membranes and with Native Membranes

The membranes were diluted to the chosen concentration with simplified ATPase assay buffer and warmed to 37 °C. The NBD amphiphiles were added from stocks in DMSO (final DMSO 1% *v/v*) by squirting the required volume into the membrane solution under gentle vortex. Fluorescence intensity was measured after 15 to 30 min incubation at 37 °C using a Cary Eclipse spectrofluorimeter (Varian, Singapore). The path length of the sample container was selected to guarantee that scattering at the excitation wavelength was smaller than 0.1 while maintaining a good signal/noise ratio (3, 5, or 10 mm).

The increase in fluorescence of the NBD amphiphiles when associated with the model membranes was well described by a simple partition (Appendix A). The low affinity of the less lipophilic amphiphiles for the native membranes required the use of high membrane concentrations, and the fluorescence was measured with the plate reader SpectraMax iD5 (Molecular Devices, Berkshire, UK), where the reading setup is not very sensitive to scatter. To guarantee that the high scatter intensity was not leading to artifacts in the measured fluorescence intensity of NBD–C4, titrations of NBD-C6 and NBD–C8 were also performed in the same plate. For these two latter amphiphiles, the results obtained for the partition coefficient were similar to those obtained for experiments with low scatter intensity. A small decrease in the fluorescence intensity from NBD–C8 was nevertheless observed at very high scatter intensities (OD ≥ 0.5), which was used to correct the signal from the less lipophilic amphiphiles, see Appendix A. The molar volume considered for the lipid was 0.8 dm^3^/mol, and 1.2 g/mL was considered for the density of the membrane proteins.

### 2.7. Bilayer Setup for Molecular Dynamics Simulations

MD simulations were used to characterize the interaction of the homologous series NBD–Cn molecules (*n* = 4, 8, 12 and 16) with a membrane model of the plasma membrane, including P–gp. The MD simulations and analysis were performed using the GROMACS 5.1.4 simulation package [50] with the Martini Coarse-Grained (CG) force field, version 2.2 [51]. Three different systems were used in this study: (i) a complex asymmetric lipid bilayer modeling plasma membranes (lipid composition adapted from Ingólfsson et al. [52] and given in Appendix A); (ii) this asymmetric membrane containing P–gp; and (iii) a simple POPC bilayer containing P–gp. The Martini model was used to set up a system containing one molecule of P–gp, 522 lipids, and 12,901 water beads for P–gp in the POPC membrane, while the complex asymmetric membrane contained one P-gp molecule, 899 lipids and 21,628 water beads.

### 2.8. Assembly of P–gp for Coarse Grain Molecular Dynamics Simulations

An atomistic configuration of the P–gp (PDB code 4M1M), with a reconstructed linker region, was converted to CG using martinize.py. The protonation states of histidine residues were defined in accordance with O’Mara and Mark [53], namely, the double protonation of His149, His583, and His1228. The protein structural conformation was maintained using the ElNeDyn22 elastic network [54] with a force constant K_SPRING_ of 500 kJ mol^−1^ nm^−2^ and a R_C_ cut-off of 0.9 nm. The assembling of the surrounding membrane and solvation with water, Na^+^, and Cl**^−^** beads in a 150 mM concentration was performed using insane.py, creating a lipid bilayer with the selected lipids in a random distribution. Both the martinize.py and insane.py scripts were obtained from the Martini webpage (http://cgmartini.nl, accessed on 26 October 2022). A snapshot of the complex membrane system containing one P–gp molecule is shown in the Appendix A.

All simulations were run using a constant number of particles under a temperature of 298 K and pressure of 1 bar and using periodic boundary conditions. Temperature and pressure control were performed using the V-rescale thermostat and Berendsen barostat, respectively, and with semi-isotropic pressure coupling. Coulomb interactions were calculated using the reaction field method with a cut-off of 1.1 nm and a dielectric constant of 15, and Lennard-Jones interactions were cut-off of at 1.2 nm. For the three bilayer systems, energy minimization was then performed using the steepest descent method, followed by two equilibration simulations of 1 ns and 10 ns, with a 2 fs and 10 fs integration step, respectively. A production run was then carried out for 10 µs with a 20 fs integration step to allow for redistribution of the lipids in the bilayer.

### 2.9. Umbrella Sampling Simulations for NBD–Cn in the Water/Complex Asymmetric Membrane

The CG topology for the NBD–Cn amphiphiles was obtained from Filipe et al. [55] NBD–C4, NBD–C8 and NBD-C12 were obtained by removing beads from the NBD–C16 alkyl chain Appendix A. The final configuration of the 10 μs unrestrained equilibration simulation of the bilayer was used as a starting point. Two NBD–Cn molecules were added, one with the center of mass (COM) of the NBD group in the center of the bilayer (*z* = 0) and the other at *z* = 4 nm (in the aqueous phase), followed by a 10 ns equilibration of the system with the z coordinate of NBD COM fixed. Initial configurations with the NBD group COM at additional *z* coordinates (−4 to 4 nm with 0.1 nm interval) were obtained by pulling the molecules along the z-axis using the pull geometry cylinder method to decrease membrane deformations [55] a pulling rate of 0.0002 nm ps^−1^ and a force constant of 500 kJ mol^−1^ nm^−2^. The initial positions of NBD–C16 are shown in the Appendix A. For the umbrella sampling simulations, a harmonic umbrella potential was applied to the NBD’s COM, with a 3000 kJ mol^−1^ nm^−2^ force constant and for 200 ns. Potential of mean force (PMF) profiles were then calculated using the WHAM method implemented in the GROMACS package [56].

### 2.10. Umbrella Sampling Simulations for NBD–Cn in the P–gp Containing Membranes

To study the transfer of the NBD–Cn amphiphiles from the lipid environment to P–gp, the reaction coordinate was defined as the distance in the xy plane between NBD COM and P–gp defined by the COM of the transmembrane (TM) region of P–gp (*d* = 0). The NBD–Cn molecules were added to the lipid bilayer at *d* = 5 nm in their equilibrium transverse position (*z*_Eq_) [57] in the inner leaflet (Appendix A). For each case, a single molecule was aligned with the P–gp gate composed of TM4 and TM6. Although the P–gp has two gates, the cleft of the TM4/6 gate was larger, which allows for the entry of larger molecules, and simulations have previously captured the entrance of a lipid through this gate [58]. A 10 ns simulation was performed with the NBD COM fixed at *d* = 5 nm to allow equilibration of the system. The NBD amphiphiles were then pulled towards the COM of the TM region, with a pulling rate of 0.0002 nm ps**^−^**^1^ and a force constant of 500 kJ mol**^−^**^1^ nm**^−^**^2^, using the pull geometry distance method. The *z* coordinate of the NBD group was not restrained, and in some situations, it moved into the water. To guarantee that initial configurations are generated for all windows (*d* = 0 to 5 nm, δ*d* = 0.1 nm), ten independent pulling simulations were performed for each molecule. The initial configurations were selected by visual inspection to guarantee that the NBD–Cn remained in the membrane and entered P–gp’s binding pocket through the TM4/6 gate. A simulation of 200 ns was then performed for each umbrella window, applying a harmonic umbrella potential to the *d* coordinate of NBD COM, with a force constant of 3000 kJ mol**^−^**^1^ nm**^−^**^2^. The sampling histograms with the probability distribution of the NBD–Cn in each umbrella window were analyzed to verify the adequacy of overlap between adjacent windows [56]. The results obtained for NBD–C8 in the complex membrane are shown in the Appendix A. PMF profiles were then calculated using the WHAM method implemented in the GROMACS package [56].

The convergence of the PMF profiles was evaluated according to the method described in Filipe et al. [55] and is shown in the Appendix A. The calculated PMF varied very significantly during the first 20 ns, and, in some cases, systematic variations are observed up to 50 ns. The final PMFs were therefore calculated from the last 150 ns of simulation. The uncertainty associated with each PMF profile was estimated as the difference between the two profiles calculated for each half (50–125 ns and 125–200 ns).

### 2.11. MD Simulations Data Analysis and Visualization

To test the effectiveness of the ElNeDyn network surrounding the P–gp, the root-mean-square deviation (RMSD) and the root-mean-square fluctuation (RMSF) of the whole protein and different domains were calculated (including the linker region), considering only the backbone beads. The martinized P–gp starting structure was used as reference. This was performed using the gmx rms and gmx rmsf GROMACS tools. The different structural motifs of the protein were defined based on the assignment of secondary structures by DSSP in the Protein Data Bank, as well as based on the attributions in Li et al. [59] and Condic-Jurkic et al. [60]. The tilt of the protein was calculated using the gmx bundle GROMACS tool by calculating the angle between the z axis of the membrane and a vector in the P–gp defined from the COM of the TM region to the COM of the nucleotide binding (NB) domains region.

The lipid distribution around P–gp may have important effects in the binding process of other molecules, such as protein substrates. Therefore, the distribution of lipids around the P–gp during the 10 µs simulation was studied by calculating the radial distribution function (RDF) and by plotting 2D-density maps using GROMACS tools gmx rdf, and gmx densmap. In this analysis, the distribution of the bead for the phosphate group (PO4 bead) of the phospholipids and the bead for the hydroxyl group (ROH bead) of cholesterol was considered.

To study the interactions of NBD–Cn with P–gp, the <*z*> distance of the NBD COM when inside P–gp was calculated both in relation to the P–gp’s TM COM and to the COM of the bilayer using the GROMACS tool gmx distance. The tool gmx select was also used to identify the P–gp residues with which the molecules interact in the *d* = 0 umbrella window simulation. For each molecule, the P–gp residues were then split into two groups: half with the higher number of interactions (above the 50th percentile) and half with the smaller number of interactions (below the 50th percentile).

For visualization of structures and trajectories, the Visual Molecular Dynamics software (University of Illinois) was used [61].

## 3. Results

### 3.1. Effect of the Amphiphiles on P–gp ATPase Activity and Competition with IAAP for Binding

The effect of the NBD amphiphiles on P–gp ATPase activity was characterized following the procedure described in the methods section. The results are shown in Figure 1, with activation being observed by the amphiphile with a smaller aliphatic chain (NBD–C4), both activation and inhibition by the amphiphile with an intermediate length of the aliphatic chain (NBD–C8) depending on its concentration, and mostly inhibition by the amphiphile with the longer aliphatic chain (NBD–LysoMPE).

The results obtained for the NBD amphiphiles agree with information in the literature for other homologous series of amphiphilic molecules, with a shift from activation towards inhibition of P–gp’s ATPase activity as the length of the aliphatic chain is increased [21,32,62,63]. This behavior has been interpreted as activation of P–gp by the binding of a single modulator molecule, as well as the uncompetitive inhibition due to binding of a second modulator molecule. The increase in the modulator hydrophobicity leads to a higher affinity for P–gp and thus decreases the modulator concentration required for the binding of the second molecule [27,45,62,63,64,65]. The parameters obtained also follow the usual interpretation, with the dissociation constant obtained for P–gp activation ($K_{1}$) being 84 µM for NBD–C4 and 17 µM for NBD–C8; and the dissociation constant for P–gp inhibition ($K_{2}$) being outside the concentration range studied for NBD–C4, and 29 µM for NBD–C8. The maximum activation ($V_{1}$) obtained was high for NBD–C4 (734% of the basal ATPase activity) and intermediate for NBD–C8 (494%).

It should, however, be noted that the uncertainty in the parameters obtained from the best fit is very large, this being particularly relevant if full inhibition cannot be assumed. A sensitivity analysis of the different parameters is performed for the case of NBD–C8 and shown in the Appendix A. It is observed that, when assuming that $K_{1}\leq K_{2}$ (Appendix A), the 75% confidence interval for $K_{1}$ is [10, 22], and that for $K_{2}$ is [22, 48]. If no constraint is imposed between the two affinity constants (Appendix A), the uncertainty in the parameters is much larger, with over one order of magnitude variation within the 75% confidence interval.

A very large uncertainty is also obtained for NBD–LysoMPE, depending strongly on the constraints imposed for the dissociation constants of the two modulator molecules. If it is assumed that $K_{1}\leq K_{2}$, the best fit of the experimental results leads to 13 µM for both dissociation constants; if no constraints exist between the dissociation constants, the best fit leads to $K_{1}$ > 5 µM (no upper limit) and $K_{2}$ < 15 µM (no lower limit). In any case, the maximal ATPase activation ($V_{1}$) obtained was always small, <126%.

To increase the confidence on the affinity of the NBD amphiphiles for P–gp, the competition with IAAP for binding to P–gp was also characterized. The results obtained are shown in Figure 2. As the concentration of the NBD amphiphiles increases, IAAP is displaced from P–gp, indicated in the figure as an increase in the inhibition of IAAP binding. This indicates that the effects observed on the ATPase activity were due to binding of the amphiphiles to P–gp’s binding pocket. As a positive control, the assay was also performed with the P–gp inhibitor Tariquidar at a concentration of 2 μM, which, as expected, showed efficient inhibition of IAAP binding, *K*_I_ = 0.2 µM [38].

The best fit considering competitive inhibition of IAAP binding (Appendix A), continuous lines, leads to a dissociation constant equal to 14, 4, and 23 µM for NBD–C4, –C8, and –LysoMPE, respectively. The prediction from the highest affinity constant obtained from the ATPase assay is also included, dashed lines, showing an underestimation of the binding affinity in the case of NBD–C4 and NBD–C8 and an overestimation in the case of NBD–LysoMPE.

Overall, the results obtained in the IAAP displacement assay suggest a similar affinity of all NBD amphiphiles for binding to P–gp. However, the ATPase assay suggests an increase in affinity as the length of the aliphatic chain of the amphiphiles increases. The discrepancy may have several origins: (i) binding of several molecules of the NBD amphiphiles in P–gp’s binding pocket. In this respect, it is relevant to note that the description of IAAP displacement assumes a single binding site with full inhibition of IAAP binding, while the binding of two amphiphiles is considered in the formalism used to describe the effects on ATPase activity. The relatively lower efficiency of IAAP displacement observed at low concentrations suggests the binding of several NBD amphiphiles with a lower efficiency in IAAP displacement for lower occupancies in P–gp’s binding pocket. (ii) Inadequate use of the total amphiphile concentration in the formalisms used to analyze the effects on ATPase activity and IAAP displacement. The amphiphiles are expected to partition efficiently to the lipid bilayer of the native membranes, and this depends on the amphiphile and on the amount of membrane. In this respect, it is important to note that the amount of membrane is 10 times higher in the IAAP assay than in the ATPase assay. (iii) Related to point (ii), the concentration of the amphiphiles in the aqueous media will be different in both assays and, depending on their partition coefficients, the relation between them may also vary.

To clarify and try to reconciliate the results obtained with both assays, we have characterized the partition of the NBD amphiphiles towards the lipid bilayer and the whole membrane of the native membranes.

### 3.2. Association of the Amphiphiles with Lipid Bilayers and P–gp Containing Native Membranes

The effects of the amphiphiles of P–gp (binding and ATPase activity) are mediated by their association with the membrane where P–gp is embedded. This mediation may be direct (through changes in the local concentration available and relative affinity for binding to P–gp), or indirect (through changes on the properties of the membrane such as fluidity or lateral pressure). Although indirect effects have been evoked, more recent reports point towards an effect on the properties of the bilayer being non-significant at the modulator concentrations where enhancement/inhibition of P–gp’s ATPase activity is observed [32,66,67]. However, the local concentration in the membrane has been estimated from the partition coefficient obtained towards simple POPC membranes or towards the air–water interface [21,32]. To clarify this question, it is necessary to obtain the partition coefficient for membranes with a representative lipid composition.

The association of the NBD-labelled amphiphiles with the membranes changes the fluorescence from the NBD group [41,42] and this has been used to obtain their partition coefficient. The results obtained are shown in Figure 3. An increase in the fluorescence intensity from the NBD moiety is observed as the concentration of native membranes increases, accompanied by a shift in the fluorescence spectra to shorter wavelengths. This behavior indicates that the NBD group is being transferred to a medium with lower polarity and/or higher viscosity, as observed when it associates with lipid bilayers [41,42].

In Figure 3A, the fluorescence intensity from NBD–C8 is represented as a function of the total concentration of protein in the native membranes. In plot B, the quantity of the membranes is expressed as the concentration of phospholipids, considering the mass ratio of phospholipids to proteins (0.57 *w/w*) and the average phospholipid molar mass of 750 g/mol obtained for these membranes [39].

The amphiphile with the longest aliphatic chain (NBD–LysoMPE) associates very efficiently with the native membranes, $\mathrm{Log}K_{P}$ = 4.34 ± 0.25. As expected, a decrease in the length of the aliphatic chain leads to a less efficient association, with $\mathrm{Log}K_{P}$ being 3.95 ± 0.14, 3.20 ± 0.26, and 2.75 ± 0.20 for NBD–C8, C6 and C4, respectively. The results obtained for the homologous series of NBD–Cn are qualitatively in agreement with what was observed for the association with POPC bilayers ($\mathrm{Log}K_{P}$ = 4.67, 3.79 and 2.99, for C8, C6, and C4, respectively) [41], although NBD–C4 shows a relatively higher affinity for the native membranes. NBD–LysoMPE has 14 carbons in the acyl chain, but its polar head group is large and negatively charged. Its partition coefficient for POPC bilayers ($\mathrm{Log}K_{P}$ = 4.2) [42] is intermediate between that of NBD–C8 and NBD-C6. The observation that NBD–C4 and NBD–LysoMPE associate more efficiently with the native membranes than predicted from their association with POPC membranes suggests that these amphiphiles interact strongly with proteins present in the native membranes. A relatively higher affinity of NBD–C4 when compared with the prediction based on NBD–Cn with longer alkyl chains has previously been observed for their binding to BSA [41], reflecting the higher relative importance of polar interactions when binding to proteins. To further understand this effect, the association of the NBD–Cn amphiphiles with pure lipid bilayers with a phospholipid composition representative of that of the native membranes was also characterized.

The lipid composition of the membranes from High-Five insect cells used in this work was characterized by HPLC-MS, allowing the quantitative characterization of the phospholipids present [39]. Phosphatidylethanolamines were the most abundant, followed by phosphatidylcholines and phosphatidylserines. The acyl composition within each class was also characterized, showing that the most abundant combination has 34 carbons and one unsaturation (most likely from a palmitoyl and an oleoyl fatty acid esterified in positions 1 and 2 of glycerol, respectively). Thus, the lipid bilayer is well modeled by a mixture of POPE, POPC, and POPS at the molar ratios 45:35:20 [39].

The relation between the partition coefficients obtained for simple bilayers prepared from POPC, and to lipid bilayers representative of the native membranes, is shown in Figure 4A. The value of LogD (Table 1) is also shown, which is an estimate of the amphiphile hydrophobicity, as evaluated from the partition between the aqueous media and octanol.

For the NBD–Cn homologous series, a linear dependence of $\mathrm{Log}K_{P}$ with the length of the alkyl chain is always observed (R^2^ > 0.999 for all). The slope is also similar, being 0.45 for LogD and 0.42 for both POPC (data taken from [41]) and PE:PC:PS 45:35:20 lipid bilayers. The intercept is, however, significantly different, reflecting the different interactions established between the amphiphiles and the distinct media. The smallest intercept is observed for CLogD (0.28), followed by $\mathrm{Log}K_{P}$ to PE:PC:PS membranes (0.89), and the largest intercept is obtained for the partition to POPC membranes (1.31). Those differences are due to the distinct solvation ability of the media and highlight the difference between hydrophobicity and lipophilicity, as well as the importance of the membrane lipid composition on the amount of amphiphile associated with the membrane.

The affinity of the NBD–Cn amphiphiles for the native membranes is compared with that for the lipid bilayer of those membranes in Figure 4B. For NBD–C8, the affinity for the native membranes (considering the volume occupied by the phospholipids in the membrane, green symbols) is the same as observed for the model membranes. In contrast, the partition coefficient of NBD–C4 to the native membranes is significantly higher than predicted and slightly higher in the case of NBD-C6. If it is assumed that the amphiphiles may interact with the phospholipids and proteins in the membrane (considering the sum of both volumes in Appendix A), then the calculated partition coefficient of NBD–C4 approaches that obtained for the model membranes, but that of NBD-C6 and NBD–C8 is underestimated. This suggests that NBD–C4 associates significantly with the proteins in the membrane, while NBD–C8 partitions mostly to the lipid pool of the native membranes.

The relative affinity of the amphiphiles for the lipid bilayer and proteins in the native membranes will be explored in the discussion section. To gain further insight into the interactions of the amphiphiles with the lipid bilayer and embedded P–gp, molecular dynamics simulations were also performed, and the results are shown in the next sections.

### 3.3. Molecular Dynamics Characterization of P–gp Embedded in Lipid Bilayers

P–gp was embedded in a simple POPC membrane and in an asymmetric lipid bilayer representative of the native membranes characterized. The RMSD values obtained for the P–gp backbone beads during a 10 μs simulations are shown in the Appendix A. The RMSD values obtained were similar for both membranes, and similar to those reported for studies with atomistic P–gp structures (namely for structures with PBD codes 4M1M and 3G5U) [60]. The RMSF values for the protein residues were also similar for both lipid bilayers, with higher flexibility in the linker region (Appendix A). The orientation of P–gp in the lipid bilayers (evaluated through the angle between the axis defined by the COM of protein TM and NB-domains, and the bilayer normal) is shown in the Appendix A for the complex asymmetric membrane. The average tilt angle is 6.3 ± 2.8° (min 0.02°, max 18.11°) in this system, whereas it is 6.8 ± 3.0° (min 0.02°, max 19.13°) for the POPC membrane. The larger tilt angle observed for the POPC membrane is due to the smaller thickness of this bilayer (3.92 ± 0.02 nm vs. 4.17 ± 0.02 nm for the asymmetric membrane) and its higher fluidity. The results obtained are in agreement with those reported for P–gp embedded in a DMPC bilayer [68].

The presence of the protein in the complex membrane influences the distribution of the distinct lipid types (Appendix A). When the average of the lipid density in the bilayer is calculated, diffuse regions of enrichment in a ring around P–gp (POPS) and near the entry gates (POPC and Cholesterol) are observed (Appendix A). Those diffuse regions become very well defined when the density is calculated, taking into account rotational orientation changes of P–gp (Appendix A), indicating that the lipid molecules maintain their position relative to P–gp during the whole 10 μs simulation. The results obtained are in agreement with previously reported results from MD and experimental observations [10,69,70].

### 3.4. MD Characterization of NBD–Cn in the Asymmetric Membrane

The PMF profiles for the transfer of the NBD–Cn amphiphiles from water to the complex asymmetric membrane (without P–gp) are shown in Figure 5. To allow a continuity in the PMF for the asymmetric system, the membrane COM was considered for the energy reference.

All amphiphiles in the homologous series interact favorably with both membrane leaflets, with a decrease in the free energy when going from water to the equilibrium position located near the glycerol group of the phospholipids (1.5 to 2.1 nm from the bilayer COM). The decrease in the free energy when associating with the membrane is larger for longer alkyl chains, as expected and previously observed both experimentally and using MD simulations for POPC membranes (Figure 5 and the literature [41,57]). The partition coefficient calculated from the free energy difference is shown in the inset for both membrane leaflets. A linear dependence is observed between Log*K*_P_ and *n*, with the slope being somewhat larger for partition to the inner leaflet (0.60 vs. 0.55), both being larger than observed experimentally for the POPC and PE:PC:PS membranes (Figure 4A). For the most hydrophobic amphiphiles (NBD-C12 and C16), the difference in the partition coefficient for the two membrane leaflets is significant, with a stronger stabilization of the amphiphile when inserted in the inner leaflet. This reflects the higher order of the outer leaflet (enriched in sphingomyelin and cholesterol), which leads to a poor solvation of solutes associated with the non-polar portion of the membrane [42,71,72,73]. The presence of cholesterol in the membranes increases the order and packing density in the non-polar portion of the membrane, but the density at the polar interface is decreased due to the very small head-group of cholesterol (Appendix A and Ref. [74]). Therefore, the membrane affinity of the amphiphiles with shorter alkyl chains (which locate mostly in the interface region) is not significantly different for both leaflets.

An estimate of the rate of NBD–Cn translocation from one membrane leaflet to the other may be obtained from the difference between the energy minimum at the equilibrium position and the maxima near the center of the bilayer (≅40 kJ/mol for the asymmetric membrane and ≅36 kJ/mol for the POPC membrane [55]). Following the methodology proposed in Filipe et al. [75], ≈100 s^−1^ is calculated for all amphiphiles in both directions in the asymmetric membrane. A quantitative agreement with experimental values is not expected, namely due to the loss of effective friction in coarse grain simulations, arising from the neglection of the fine-grained degrees of freedom [76]. The results obtained do, however, predict that NBD–Cn equilibrates fast between the leaflets of the native membranes, as observed experimentally in POPC membranes [77].

### 3.5. MD Characterization of the Transfer of NBD–Cn from the Lipid to P–gp

The PMF profiles for the transfer of the NBD–Cn from their equilibrium position in the lipid bilayer towards P–gp were obtained for the simple (POPC only) and for the complex asymmetric membrane using the radial distance between the COM of the amphiphile polar NBD group and that of the P–gp TM region as reaction coordinate. The PMFs are shown in Figure 6, and their convergence is shown in the Appendix A.

When the bulk membrane is considered for the reference in the PMF profile (plots A,B), it is clear that the energy of all NBD–Cn amphiphiles in the lipid membrane remains unchanged when going from bulk down to about 3 nm from the P–gp COM. At distances between 2 and 3 nm, the free energy of the amphiphiles shows some fluctuations but does not deviate significantly from the value in the bulk lipid bilayer. At distances below 2 nm, an increase in free energy is observed in the asymmetric membrane, as the amphiphiles approach the outer surface of the protein and enter P–gp’s gate region. A small decrease in the free energy is, however, observed for all amphiphiles (except NBD–C16) in the POPC membrane when approaching the outer surface of P–gp, followed by a free energy increase as they enter the gate region. After crossing P–gp’s gate, the free energy continues to increase for all amphiphiles, except for NBD–C4, which shows stabilization when inside P–gp’s binding pocket.

The distinct behavior observed for NBD–C4 is very clearly observed when considering the center of P–gp as reference (plots C,D). While the energy of the amphiphiles with longer alkyl chains continuously decreases as they move from the protein towards the lipid membrane, NBD–C4 interacts with similar affinity with both P–gp and the lipid membrane. This is in very good agreement with the experimental results obtained for the association of the amphiphiles with the native membranes (containing membrane proteins), Figure 4B, shows that the affinity of NBD–C4 for P-gp is relatively high, while the longer alkyl chain amphiphiles interact mostly with the lipid portion of the membranes.

### 3.6. Details of the Interaction of the NBD–Cn Amphiphiles with P–gp

In the simulations from which the PMFs shown in Figure 6 where calculated, only the distance to the TM COM of the P–gp was restrained. The amphiphiles were free regarding their transverse position, both when in the membrane and when interacting with P–gp. The transverse position at the beginning of the simulation was generated from pulling the amphiphile from its equilibrium position in the inner leaflet of the membrane towards the COM of the TM region of P–gp oriented with P–gp’s gate between TM4 and TM6. During the simulations at each window, the transverse location of the NBD group varied depending on the interactions established (Figure 7 for NBD–C4 and S17 to S20 for all NBD–Cn). The transverse position was calculated relative to the P–gp TM COM and converted into the position relative to the bilayer center through the average of the relative position of P–gp TM COM (1.10 ± 0.13 nm). This conversion allows for a better comparison with the PMF for the amphiphile in the membrane, Figure 5. For *d* ≥ 2.3 nm (in the lipid bilayer), the transverse position shows a broad distribution centered at around 2 nm (Figure 7, plots A,B). This broad distribution reflects variation in the local transverse position of the NBD group and also undulations in the membrane. When NBD–C4 is able to interact with residues in the outer surface of P–gp (1.1 nm ≤ *d* ≤ 2.2 nm, plots C,D), its transverse position depends on the interactions established and changes over time, leading to multimodal distributions. When crossing the gate (*d* ≅ 1 nm, plot E), the fluctuations on the transverse location of NBD–C4 usually decrease, leading to sharp distributions. Finally, when the NBD group of NBD–C4 is inside the binding pocket (*d* ≤ 0.6 nm, plots F–H), multimodal distributions are observed again indicating interaction with different regions of P–gp.

The multimodal distribution of the transverse location of the NBD group shows that interaction with different regions of the protein is being sampled. The P–gp’s residues with which the different amphiphiles interact when restrained at the center of the binding pocket (*d* = 0) are shown in Figure 8 and in the Appendix A. Residues from the transmembrane helices TM3, 10, 11 and 12 establish interactions with all NBD–Cn amphiphiles, while residues from TM6 are also involved in interactions with NBD–C4 and C12, and residues from TM4 interact with NBD–C8. The residues involved in the interactions are typically common to those observed for most P–gp substrates [29,68,78,79], including hydrophobic (such as Leu and Phe) and also polar amino acids (especially Asp and Lys), see Appendix A for details.

It should be noted that, in spite of the extensive simulations performed, the systems studied are still not fully described. The multimodal distributions in the transverse position of the NBD group when interacting with P–gp indicate sampling of the surface of P–gp’s binding pocket. However, the residence time in each position is high, and the 200 ns simulation time does not allow for a complete sampling. It is also observed that the transverse position, when interacting with P–gp, is strongly dependent on the initial position generated from the pulling, leading to NBD–C4 and NBD-C12 located at *z* ≅ 2 nm when inside P–gp’s binding pocket, while NBD–C8 and NBD–C16 are located at *z* ≅ 3 nm (Appendix A). To gain confidence in the preferential location and interactions established between the distinct NBD–Cn and P–gp, a significantly better sampling would be necessary. This cannot feasibly be achieved through an increase in the simulation time nor with independent simulations. New methodologies with improved sampling are required.

In spite of the limitations, important information regarding the energetics and some structural details of the interactions between the NBD–Cn and P–gp was obtained. A more detailed analysis will be provided in the discussion, in conjunction with the experimental results obtained in this work.

## 4. Discussion

All amphiphiles studied in this work are able to inhibit IAAP binding to P–gp and to change P–gp’s ATPase activity. The observation that the NBD amphiphiles are able to displace IAAP clearly indicates that they bind to the drug-binding pocket of P–gp as the IAAP is a transported substrate. The ATPase activity assay gives information regarding the effect of the amphiphiles on the conformational dynamics of P–gp and is usually taken as evidence for transport by P–gp [80]. The results would thus indicate that the NBD–Cn amphiphiles are transported by P–gp, while NBD–LysoMPE essentially inhibits P–gp conformational dynamics. An inverse relation between the binding affinity for P–gp and the effect on ATPase activity has been previously observed, with moderate binding leading to an enhancement of ATPase activity, while strong binding leads to P–gp inhibition [21]. A similar behavior is observed in this work for the NBD–Cn when considering its total concentration and evaluating the binding affinity from the effects on P–gp’s ATPase activity. However, when binding is evaluated from the extent of IAAP displacement, a similar affinity is obtained for all the amphiphiles studied. This inconsistency may be due to differences in the two properties (binding and effects on ATPase activity), or it reflects the limitations and inadequacy of the models used to analyze the results.

The set of experiments performed in this work, together with the extensive knowledge on the properties obtained previously for those amphiphiles [41,42,71,77,81,82,83], allows an integrated interpretation of the results, possibly elucidating the discrepancies observed between the two sets of experiments. These are discussed below.

### 4.1. Effective Concentration of the Amphiphiles

A major limitation of the models used to analyze the effects on P–gp’s ATPase activity and IAAP displacement is the use of global concentrations. Interaction of the amphiphiles with P–gp is considered to occur from the membrane [8,9,10] and, therefore, what is relevant is the amphiphile concentration in the lipid bilayer where P–gp is embedded. The importance of using local concentration in the analysis of the interaction of small molecules with membrane proteins has been pointed out before (e.g., [33,35,37,46,84,85]), but, unfortunately, it is still overlooked. The local concentrations may be calculated from the partition coefficient of the amphiphiles between the aqueous media and the model membranes with the relevant lipid composition. The results obtained (Figure 4A) show that NBD–C4 is associated with the lipid bilayer with the lowest affinity and NBD–C8 is associated with the highest affinity. An intermediate affinity is obtained for NBD–LysoMPE in spite of its longer aliphatic chain. This is due to its bigger and much polar (charged) head group. From the partition coefficients one may calculate the local concentration of amphiphile in the membrane for a given total concentration of amphiphile. At the concentration of native membranes used in the ATPase assays (0.107 mg of protein per mL), 2% of the NBD–C4 is associated with the lipid bilayer of the native membrane (see Appendix A for details). As expected, for the more lipophilic amphiphiles, the fraction associated with the lipid bilayer increases, being 53% for NBD–C8 and 13% for NBD–LysoMPE. In the IAAP displacement experiments, the amount of native membranes is almost 10 times higher (1 mg of protein per mL), leading to 18% of NBD–C4 associated with the membranes, and 91% and 58% for the case of NBD–C8 and NBD–LysoMPE, respectively.

The amphiphiles with higher partition coefficients towards the membrane, have a higher concentration in the membrane from where they interact with P–gp. Therefore, for those amphiphiles the apparent affinity for P–gp may be larger even for a weaker intrinsic affinity for P–gp. When the amount of membrane is changed the relative effective concentration of the amphiphiles is altered. Therefore, the relative apparent affinities of the amphiphiles for P–gp are different when obtained from the global amphiphile concentration in the ATPase and IAAP displacement assays.

From the effective concentration of amphiphile in the membrane it is relevant to calculate the ratio of phospholipid to amphiphile (PL:NBD), to evaluate possible changes in the membrane properties as the reason behind the distinct behavior observed between the amphiphiles and assays. For the conditions of the ATPase assay and a total concentration of 60 µM NBD–C4, 1.4 µM is associated with the membrane corresponding to PL:NBD equal to 58. When the membrane concentration is increased 10-fold in the IAAP assay, the amount of NBD–C4 associated with the membrane increases to 11 µM, but the value of PL:NBD increases to 70. The corresponding calculations for the case of NBD–C8 at the highest concentration used (50 µM) lead to a PL:NBD of 3 in the ATPase assay and 17 in the IAAP assay; and to PL:NBD of 13 or 26 for NBD–LysoMPE in the ATPase and IAAP assay respectively. It has been previously shown that for small and uncharged or with a single charge amphiphiles (at high ionic strength), the membrane properties are not significantly perturbed for PL:NBD values down to 20 [49] which is the case observed in the IAAP for all amphiphiles and concentrations, but not for NBD–C8 or NBD–LysoMPE on P–gp ATPase activity assay.

It should be noted that although the value calculated above for PL:NBD is higher (lower local concentration) for NBD–LysoMPE than for NBD–C8, this parameter may actually be similar for both due to the distinct rate of translocation between the two membrane leaflets. The rate of NBD–C8 translocation is fast, and equilibration between both membrane leaflets is expected to occur within a few seconds (Section 3.4 and [77]). In contrast, the negative charge on the phosphate of NBD–LysoMPE leads to a slow rate of translocation with a characteristic time of several hours [81]. The amphiphile is therefore not able to fully equilibrate between the two membrane leaflets during the time of the ATPase assay, leading to an asymmetric distribution with a higher local concentration in the outer leaflet. This has two important consequences on the effect of this amphiphile on P–gp ATPase activity: (i) the local concentration on the membrane leaflet that has access to P–gp binding pocket (the outer leaflet for the inside-out vesicles used in the assays) is larger than that predicted at full equilibration; and (ii) the asymmetric distribution changes the membrane local curvature and lateral pressure profile, with possible effects on membrane permeability and on the conformational dynamics of membrane proteins [69,72,86,87,88,89,90].

Another concern regarding the formalisms commonly used to analyze the amphiphiles interaction with P–gp (Appendix A) is the consideration of the total ligand concentration instead of the concentration of free ligand, that is, the assumption that the amphiphile is in large excess relative to the protein. We have recently reviewed the limitations of this approach and showed that, for high and moderate affinities (dissociation constants in the µM range or lower), significant deviations are easily observed between the true and the estimated affinity [40]. In the specific conditions of this work, the concentration of P–gp (6 and 60 nM in the ATPase and IAAP displacement assays, respectively) is, in fact, much lower than the total concentration of the NBD–amphiphiles. However, their concentration in the aqueous phase may become comparable to that of P–gp due to sequestration by the lipid membrane.

### 4.2. Concentration of the Amphiphiles in the Aqueous Phase

One important aspect which is usually overlooked in the studies of interactions with P–gp is whether the molecule of interest is in its monomeric state or aggregated. Some studies with detergents have included information regarding the formation of micelles and concluded that a complex effect on the ATPase activity (enhancement and inhibition) is observed even for concentrations below their CMC [21,32,91]. However, small aggregates are formed at concentrations several orders of magnitude smaller than the concentration at which micelles are detectable (their CMC) [92,93,94,95]. The aggregation behavior of the NBD amphiphiles used in this study was characterized by changes in the fluorescence properties of the NBD group and is, therefore, sensitive to the formation of small aggregates including dimers. The critical aggregation concentrations (CAC) at 25 °C are 1, 4, and 250 μM for NBD–C8, NBD–LysoMPE, and NBD–C4, respectively [41,42]. In the presence of co-solvents that disrupt the structure of water and thus decrease the hydrophobic effect (e.g., 1 M trehalose), the CAC of NBD–C4 is not significantly affected, but that of NBD–C8 is increased by 4 fold and that of NBD-C10 is increased by 6 fold [82]. An increase to at least 24 µM is thus predicted for NBD–LysoMPE.

From the above, it is clear that NBD–C4 in the aqueous media is always in its monomeric state, but to evaluate possible aggregation of NBD–C8 and NBD–LysoMPE in the aqueous phase, it is necessary to consider their association with the lipid membranes. Considering the partition coefficients obtained for the native membranes, and a total concentration of the amphiphiles equal to 50 µM, one calculates (see Appendix A for details) that the concentrations of NBD–C8 and NBD–LysoMPE in the aqueous phase are decreased to 21 µM and 12 µM at the conditions of the ATPase assay, and to 4 µM and 2 µM at the conditions of the IAAP assays, respectively. The concentrations of the amphiphiles in the aqueous phase are close to or lower than the amphiphiles CAC in aqueous medium at 25 °C in the IAAP assay and no aggregation is expected. However, in the ATPase assay significant aggregation of NBD-C8 in the aqueous phase may occur (in spite of the higher solubility expected at 37 °C and 20 to 200 mM glycerol). The inhibition of the P–gp ATPase activity by NBD-C8 could, therefore, be partially due to the interaction of small aggregates of the amphiphile formed in the aqueous medium. Although the currently accepted model for the interaction between amphiphiles and P–gp considers that it is the amphiphile in the membrane that associates with P–gp, it has been recently suggested, based on results from MD simulations, that small aggregates of detergents could interact with P–gp from the aqueous media [66].

### 4.3. Relative Affinity of the Amphiphiles for the Lipid Bilayer and for P–gp

From the partition coefficient obtained for the native membranes, $K_{P}^{W\to M}$, and that for the model lipid bilayers, $K_{P}^{W\to \mathrm{Lb}}$, one can calculate the partition coefficient between the aqueous phase and the proteins in the membrane, $K_{P}^{W\to P}$, and from the lipid bilayer to the proteins, $K_{P}^{\mathrm{Lb}\to P}$, Equations (1) and (2), [40].
(1)$$K_{P}^{W\to M}\;V_{M}=K_{P}^{W\to P}V_{P}+K_{P}^{W\to \mathrm{Lb}}\;V_{\mathrm{Lb}}$$
(2)$$K_{P}^{\mathrm{Lb}\to P}=\frac{K_{P}^{W\to P}}{K_{P}^{W\to \mathrm{Lb}}}$$

In Equation (1) it is necessary to include the volume occupied by the proteins, and the values obtained will, therefore, depend on whether it is considered that the amphiphile may interact with all proteins in the membrane or with P–gp only. The values obtained in both scenarios are provided in Table 2.

**Table 2 pharmaceutics-15-00174-t002:** Partition coefficient of the amphiphiles between the distinct media **^1^**.

	$K_{P}^{W\to M}$ ** ^2^ **	$K_{P}^{W\to \mathrm{Lb}}$	$K_{P}^{W\to P}$	$K_{P}^{\mathrm{Lb}\to P}$
Binding to the protein P–gp only				
NBD–C4	1.4 × 10^3^	3.9 × 10^2^	6.9 × 10^4^	176
NBD-C6	3.9 × 10^3^	2.4 × 10^3^	1.0 × 10^5^	43
NBD–C8	2.2 × 10^4^	1.9 × 10^4^	2.3 × 10^5^	12
Binding to all membrane proteins with equal affinity				
NBD–C4	5.7 × 10^2^	3.9 × 10^2^	6.9 × 10^2^	1.8
NBD-C6	1.6 × 10^3^	2.4 × 10^3^	1.0 × 10^3^	0.43
NBD–C8	9.0 × 10^3^	1.9 × 10^4^	2.3 × 10^3^	0.12

^1^ The partition coefficient followed a LogNormal distribution (as expected [96]) and, therefore, the reported value was calculated from the average of Log(*K*_P_). The uncertainty in Log(*K*_P_) was between 0.04 and 0.26, being typically 0.2. ^2^ Obtained from the experimentally observed partition of the amphiphiles to the native membrane considering the phospholipid bilayer and P–gp, or the phospholipid bilayer and all proteins when calculating the volume of the membrane.

It is observed that the relative affinity of the amphiphiles from the aqueous medium towards the proteins in the membrane

$\left(K_{P}^{W\to P}\right)$ increases as the amphiphile lipophilicity increases (from *n* = 4 to 8). However, this does not accompany the increase observed in their affinity for the lipid bilayer $\left(K_{P}^{W\to \mathrm{Lb}}\right)$. Therefore, a decrease is observed in the affinity for the membrane protein, relative to that of the lipid bilayer $\left(K_{P}^{\mathrm{Lb}\to P}\right)$. This is in agreement with the results obtained by MD simulations (Figure 8).

At low amphiphile concentrations, away from saturation of the lipid bilayer and proteins, the fraction of amphiphile associated with each medium may be calculated from the respective partition coefficient and volume. It was found that 72% of NBD–C4 in the membrane is associated with proteins, and this decreases to 15% for NBD–C8.

### 4.4. Number of Binding Sites in P–gp’s Binding Pocket

There is strong evidence that P–gp’s binding pocket may bind several molecules simultaneously, both indirect evidence from competition or binding experiments (e.g., [25,26,27,28,29,31,34,97]), and, more directly, from P–gp crystalized in the presence of their substrates [98]. The results obtained in this work also point towards the existence of several overlapping binding sites: (i) several locations of the amphiphiles in P–gp’s binding pocket have been observed by MD simulations (Figure 7, Figure 8 and Appendix A); and (ii) evidence for a less-efficient IAAP displacement at smaller concentrations of amphiphile when compared to intermediate concentrations (Figure 2).

However, the formalisms usually considered for the analysis of the results are not able to provide quantitative information for the number of molecules that may bind to P–gp’s binding pocket. Additionally, in spite of the strong evidence, the effect of modulators on P–gp ATPase activity or ligand displacement are usually analyzed considering the binding of a single ligand molecule. For the case of modulators of ATPase activity, the equation used includes both enhancement and inhibition, Appendix A, depending on modulator concentration. However, the derivation of this equation cannot be found in the literature, and the physical meaning of the parameters (e.g., in terms of microscopic or macroscopic, global of stepwise binding constants [99], etc.) is not clear.

We have recently shown that the partition coefficient obtained at low ligand concentration is related to the macroscopic binding constant of the first ligand $\left(\beta_{1}\right)$, Equation (3) [40], where $\bar{V_{P}}$ is the molar volume of the binding agent.
(3)$$\beta_{1}=\sum_{i=1}^{n}K_{b}^{i}\;=K_{P}^{W\to P}\bar{V_{P}}$$

If it is considered that all binding sites are equal $\left(K_{b}^{i}=K_{b}\;\forall \;i\right)$, the number of binding sites (n) may be obtained from the comparison between the partition coefficient and the binding affinity, Equation (4). Given the required approximation of equal binding sites, the number of binding sites is replaced by # to call to attention that it is only an estimate of the true parameter, n. In the analysis below, the molar volume of P–gp was calculated from its molar mass, assuming a density of 1.2 g/mL, leading to $\bar{V_{P}}$ = 142 dm^3^/mol.
(4)$$\#\;=\frac{K_{P}^{W\to P}\bar{V_{P}}}{K_{b}}=K_{P}^{W\to P}\bar{V_{P}}\;K_{d}$$

Using this formalism, it is possible to obtain an estimate of the number of amphiphiles that may be accommodated in P–gp’s binding pocket. However, the dissociation constant used in Equation (4) must be obtained taking into account the local concentration of ligand in the aqueous phase, not its total concentration. This concentration is affected by sequestration in the lipid bilayer, which can be accounted for using the partition coefficient obtained for the model membranes. The concentration of ligand in the aqueous medium is also influenced by binding to the protein, which depends on the number of binding sites considered. Thus, to estimate the number of binding sites using Equation (4) it is necessary to use a formalism that explicitly includes the number of binding sites. This will be performed in the next sections.

Before re-analyzing the experimental results for the ATPase activity and IAAP displacement, it is important to discuss the physical meaning of the association constants obtained from the ligand in the aqueous phase towards the protein in the membrane. A schematic description of P–gp in the inside-out vesicles prepared from the native membranes is presented in Figure 9.

The ligands interact with P–gp from the lipid bilayer (dependent on $K_{P}^{\mathrm{Lb}\to P}$), this being dependent on the partition from the aqueous medium towards the lipid bilayer ($K_{P}^{W\to \mathrm{Lb}}$). This is a composite process that cannot be easily described when analyzing the results for the effects of the ligands in P–gp. The equilibrium between the aqueous medium and P–gp ($K_{P}^{W\to P}$) closes the thermodynamic cycle and thus reflects the other two equilibria in the cycle. Even if ligand interaction with P–gp does not occur from the aqueous medium, the equilibrium constant (or partition coefficient) may be calculated from the alternative path in the cycle and reflects accurately the affinity of the ligand for the membrane protein.

### 4.5. Re-Analysis of the Effects on ATPase Activity

For the correct analysis of the results, the distribution of the amphiphiles in the distinct environments (aqueous media, lipid bilayer, and P–gp) must be considered. In addition, excess ligand should not be assumed. When considering that P–gp has several binding sites, the kinetic scheme that describes P–gp saturation with ligand is given by Equation (5). For details, see, e.g., references [40,99,100].
(5)$$\begin{array}{l} L_{W}\overset{K_{P}^{W\to \mathrm{Lb}}}{↔}L_{\mathrm{Lb}}\; [L_{\mathrm{Lb}}]=K_{P}^{W\to \mathrm{Lb}} [L_{W}]\;V_{\mathrm{Lb}}/V_{W}\\ L_{W}+P\overset{\beta_{1}}{↔}PL_{1}\; [PL_{1}]=\beta_{1} [P] [L_{W}]\\ i\;L_{W}+P\overset{\beta_{i}}{↔}PL_{i}\; [PL_{i}]=\beta_{i} [P]{ [L_{W}]}^{i}\;\\ n\;L_{W}+P\overset{\beta_{n}}{↔}PL_{n}\; [PL_{n}]=\beta_{n} [P]{ [L_{W}]}^{n} \end{array}$$

In the above equations the concentration of the ligand in the distinct environments corresponds to the amount in that environment divided by the total volume of the solution. The macroscopic association constants ($\beta_{i}$) are related with the macroscopic stepwise association constants ($K_{i}$) by:(6)$$\beta_{i}=\prod_{j=1}^{i}K_{j}$$

For the case of equal and independent binding sites, $K_{i}$ is related with the microscopic association constant for each binding site $\left(K_{b}\right)$ by:(7)$$K_{i}=\frac{n-i+1}{i}K_{b}$$

To calculate the concentration of each species it is necessary to solve the set of equations
(8)$$\begin{array}{cc} [L_{T}] & = [L_{W}]+ [L_{\mathrm{Lb}}]+\sum_{i=1}^{n}i [PL_{i}]\\ & = [L_{W}]+ [L_{\mathrm{Lb}}]+\sum_{i=1}^{n}i\;\beta_{i}\; [P]{ [L_{W}]}^{i}\\ [P_{T}] & = [P]+\sum_{i=1}^{n} [PL_{i}]= [P]+\sum_{i=1}^{n}\beta_{i}\; [P]{ [L_{W}]}^{i} \end{array}$$

For n = 1, this leads to a quadratic equation that may easily be solved. However, for n > 1, the concentration of free ligand in solution must be obtained using numerical methods [40,100]. The concentration of free protein may be calculated from the bottom line in Equation (8), and that of all other species may then be calculated from the concentration of free ligand and protein using Equation (5).

The overall ATPase activity at each ligand concentration is calculated from the concentration of each protein species $\left( [PL_{i}]\right)$ and the corresponding ATPase activity $\left(V_{i}\right)$,
(9)$$V=V_{0}+\sum_{i=1}^{n} [PL_{i}]\;V_{i}$$

This formalism was used to describe the effect of the NBD amphiphiles on P–gp’s ATPase activity, considering a distinct ATPase activity for P–gp with a different number of amphiphile molecules in the binding pocket. The results are shown in Figure 10.

It is observed that the quality of the best fit improves very significantly when the number of binding sites is increased from one to three, and small improvements are observed for larger values of n. More importantly, the value of $K_{P}^{W\to P}$ calculated from $\beta_{1}$ increases with n, approaching the value calculated from the partition to the lipid bilayer and to the whole native membranes (Table 2). This strongly suggests that P–gp accommodates at least three molecules of each NBD amphiphile in its binding pocket. The value obtained for the microscopic dissociation constant from each binding site is also shown in Figure 10 plots B,D,F. It is observed that, as n increases (accompanied by an improvement in the quality of the best fit), $K_{d}$ becomes similar for all amphiphiles (between 3 and 9 µM). This suggests that the major difference between the amphiphiles is on their effect on P–gp’s ATPase activity, not so much on their affinity for P–gp. The effects on P-gp are reflected on the values of the ATPase activity at the different protein saturations (Appendix A). It is observed that, for low protein saturation ($\bar{n_{P}}$ = 1), the enhancement of ATPase activity is small for all amphiphiles studied, increasing as the protein saturation increases. For NBD–C4, an enhancement of the ATPase activity is obtained up to $\bar{n_{P}}$ = 5. However, for NBD–C8 and NBD–LysoMPE, a maximal activation is observed for $\bar{n_{P}}$ = 2 or 3, and larger occupancy numbers lead to inhibition of P–gp’s ATPase activity.

### 4.6. Re-Analysis of the Effects on IAAP Displacement

In this assay, in addition to the binding of the NBD amphiphiles to P–gp binding pocket, it is also necessary to consider the binding of IAAP. The concentration of IAAP used in the assay (5 nM) is several orders of magnitude smaller than that of the amphiphiles in study, and it is also smaller than the total concentration of P–gp (59 nM). Therefore, Equation (8) may be used to calculate the concentration of free ligand and protein. The fraction of IAAP bound to P–gp may then be calculated from Equation (10), where $[P^{*}]$ is the concentration of P–gp available for IAAP binding.
(10)$$\begin{array}{l} \frac{ [I_{P}]}{ [I_{T}]}=\frac{-b-\sqrt{b^{2}-4ac}}{2a}\\ a=1\;;\;b=-\left(1+\frac{K_{d}^{\mathrm{IAAP}}}{ [I_{T}]}+\frac{ [P^{*}]}{ [I_{T}]}\right)\;;\;c=\frac{ [P^{*}]}{ [I_{T}]} \end{array}$$

The number of binding sites in P–gp binding pocket was varied from 1 to 5, and three situations were considered in the analysis: (i) full inhibition of IAAP binding when at least one NBD amphiphile is in the P–gp binding pocket (i≥1), corresponding to $[P^{*}]= [P]$; (ii) inhibition of IAAP binding only when P–gp is fully saturated (i=n), corresponding to $[P^{*}]= [P]+\sum_{i=1}^{n-1} [PL_{i}]$; and (iii) an intermediate situation with IAAP displacement for i≥n−1, corresponding to $[P^{*}]= [P]+\sum_{i=1}^{n-2} [PL_{i}]$.

The displacement of IAAP from P–gp binding pocket was calculated from Equation (11), where ${ [I_{P}]}_{0}$ is the concentration of IAAP bound to P–gp in the absence of the NBD amphiphiles.
(11)$$\mathrm{IAAP}\;\mathrm{displacement}\;\left(\%\right)=100\frac{{ [I_{P}]}_{0}- [I_{P}]}{{ [I_{P}]}_{0}}$$

The best fit and parameters for the case of intermediate efficiency in IAAP displacement (i≥n−1) are shown in Figure 11, those obtained for the efficiencies of the two extreme situations (i≥1 or i=n) are shown in the Appendix A.

It is observed that the quality of the best fit improves as n is increased from 1 to 5 for all NBD amphiphiles. This is particularly evident for NBD–LysoMPE, where the accentuated cooperative behavior is only captured when a large number of binding sites is considered. As expected, the partition coefficient calculated from $\beta_{1}$ increases with. For the case of NBD–C4, an excellent agreement is obtained between the two estimates for n = 5, while the results obtained for NBD–C8 suggest a smaller number of molecules required to displace IAAP or a lower number of NBD molecules that may be accommodated in P–gp binding pocket (n = 3 or 4). The opposite situation is obtained for NBD–LysoMPE, with the suggestion of a larger number of molecules being required to displace IAAP. The strong evidence for binding of several NBD amphiphiles in P–gp’s binding pocket may be due to their small size when compared to common P–gp substrates. Support for a size-dependent behavior may be found in the literature, where stipiamide monomers show a low efficiency at small concentrations and a cooperative behavior in the inhibition of IAAP binding to P–gp, while covalent dimers of stipiamide are more efficient and show a simple dependence with concentration [101] or a more efficient inhibition of P–gp transport activity by bisbenzylisoquinoline alkaloids [102]. Further evidence for the binding of several molecules in P–gp’s binding pocket has also been obtained from docking studies (e.g., [9,29,79,103,104,105,106,107,108]).

The consensus value obtained for the microscopic dissociation constant (considering the quality of the best fit and the agreement between the two estimates for the partition coefficient) was very similar for all NBD amphiphiles, varying between 2 and 8 µM. Those estimates are in excellent agreement with the results obtained for the effects of the amphiphiles on ATPase activity. This suggests that the interactions between the amphiphiles and P–gp binding pocket are mostly established by the polar region of the amphiphiles. This has been observed before for several detergents [63,109]. However, because the effects on P–gp’s properties were being accessed considering the overall concentration of the amphiphile, the apparent binding affinity for P–gp was influenced by the length of the non-polar tail, thus suggesting an increased affinity for P–gp with the increase in lipophilicity [32,63]. In this work, the local concentrations relevant to each equilibrium constant are used, clearly showing that the affinity for P–gp is not significantly affected by the length of the aliphatic chain. Thus, the distinct effect observed on ATPase activity (enhancement or inhibition) cannot be due to distinct occupancy numbers on P–gp. Instead, it should be due to distinct effects on P–gp properties, namely, on its conformational dynamics.

### 4.7. Hypothesis for the Mechanism of Inhibition of P–gp ATPase Activity

In agreement with the observation that P–gp inhibitors are on average more hydrophobic than P–gp substrates [20,21,22,32], several hypotheses have been proposed for the mechanism of inhibition of P–gp’s ATPase activity by some molecules. A membrane-mediated mechanism has been proposed, where the accumulation of the inhibitor in the membrane would lead to membrane perturbation [110,111,112]. It has also been suggested that inhibition is a direct consequence of a very high affinity for P–gp [21]. More recently, molecular dynamics simulations and docking studies have shown that large molecules with several aromatic moieties bind to P–gp at a deeper position (the M site) [103,104]. In this position, the ligand may establish cross interactions between the TMs of both P–gp halves, which may impair conformational changes required for P–gp’s catalytic cycle.

The membrane mediated hypothesis has been discarded as a general mechanism because several studies indicate that no significant membrane perturbation is observed at the relevant ligand concentrations [28,32,66,67]. The results obtained in this work support this observation. In the ATPase activity assay, NBD–C8 and NBD–LysoMPE may reach very high local concentrations in the membrane (less than 10 phospholipid molecules per amphiphile in the membrane at the higher amphiphile concentration used), which could lead to changes in the membrane properties. However, inhibition of ATPase activity is observed even at relatively low local concentration of the amphiphiles ([NBD–LysoMPE] ≥ 5 µM corresponding to PL:NBD ratios ≤ 126) pointing towards additional mechanisms for the inhibition.

The results obtained with the analysis of the effects on ATPase activity considering the local concentration of the amphiphiles in the distinct environments and the possible binding of several molecules in P–gp’s binding pocket suggest that inhibition may be due to the presence of a large number of amphiphiles that may difficult P–gp’s conformational dynamics. This has in fact been proposed as a possible inhibition mechanism [113,114]. An important test for this hypothesis would be to perform the ATPase assays with purified P–gp at different ratios of P–gp/lipid. The modulator will be distributed between the different media, aqueous, lipid environment and P–gp. At sufficiently low P–gp/lipid ratios, most of the amphiphile molecules will be in the lipid phase, and none of the two media would be saturated. A low occupancy number would thus be observed in P–gp’s binding pocket. However, as the volume of the lipid phase decreases (leading to a higher P–gp/lipid ratio), sequestration of the amphiphile in the lipid phase would be less significant. In this case, at the same total concentration of protein, P–gp would become saturated with modulator at lower total modulator concentrations. In light of this hypothesis, it is interesting to note that tariquidar inhibits P–gp’s ATPase activity when the protein is at a ratio of about 1% *w/w* (native membranes and reconstituted in proteoliposomes or nanodiscs), while it leads to ATPase activation for P–gp in n-dodecyl maltoside (DDM) micelles when the protein is at a ratio of 0.1% [38]. The lower relative abundance of P–gp when in the DDM micelles could also explain the much lower efficiency of tariquidar in the inhibition of IAAP binding to P–gp [38]. Notwithstanding the effects of different lipids on P–gp properties (in particular the distinct effects of a lipid bilayer or a small micelle on P–gp’s conformational dynamics), the larger relative volume of the lipid phase when P–gp is in the DDM micelles will sequester a larger fraction of modulator leading to a lower saturation of P–gp and a lower apparent affinity [108]. It is also relevant to note that a higher apparent affinity (when compared to P–gp) was observed for several molecules towards the efflux transporter ABCG2 [21] and a P–gp mutant [115], when these transporters were at a higher fraction in the membrane.

The results obtained in this study also show that inhibition of P–gp’s ATPase activity depends on the properties of the amphiphile, with NBD–C4 leading to activation even when five amphiphiles are bound, while a smaller number of molecules would already lead to inhibition in the case of NBD–C8 and NBD–LysoMPE.

The MD simulations shed some light regarding a possible additional mechanism of inhibition which depends strongly on the properties of the amphiphile. In Figure 12, a snapshot of NBD–C16 (with a long 16 carbons alkyl tail attached to the polar NBD group) is presented for a distance of 0.5 nm between the center of mass (COM) of NBD and that of P–gp’s transmembrane helices in the binding pocket. It is observed that the polar NBD group has passed the entry gate and is inside P–gp’s binding pocket. However, the long alkyl chain is extended towards outside the entry gate, maintaining interactions with the lipid bilayer. A similar orientation, albeit less pronounced, is also observed for NBD-C12 and for NBD–C8, but not for the amphiphile with the smaller alkyl chain (NBD–C4), Appendix A.

The orientation of the long aliphatic tail of surfactants towards the lipid membrane is well documented in the literature [32,66,116,117]. However, the assumption that inhibition is due to a single molecule bound to the so-called inhibitor site, masked this behavior as a possible mechanism of inhibition. A large number of amphiphiles with bulky hydrophobic groups bound to P–gp’s binding pocket would create a ring at the lipid bilayer/P–gp interface that could prevent the conformational changes required for P–gp ATPase activity. This would explain the inhibition of the ATPase activity of P–gp (and other efflux transporters) in the presence of detergents or drugs with long hydrophobic tails [32,66,116,118].

## 5. Conclusions

The comprehensive work performed in this study for the interaction of several amphiphiles with P–gp (effects on ATPase activity and IAAP displacement), and with the membranes where P–gp is embedded (native membranes and pure lipid bilayers), complemented with MD simulations and previous studies on the solubility of the amphiphiles in aqueous media and on the kinetics and equilibrium of their interaction with lipid bilayers, [41,42,71,77,81,82] allows an integrated interpretation of the results and several important conclusions.

It is shown that, to interpret the results from effects on P–gp’s ATPase activity and/or displacement of known P–gp ligands in term of P–gp specificity, it is necessary to consider the distribution of the ligand in the distinct media (aqueous, lipid bilayer, and P–gp). The importance of ligand sequestration is widely recognized [33,35,37,46,84,85], but the formalisms used to characterize binding to P–gp conflictingly consider the total concentration of ligand. This leads to apparent affinities that preclude the establishment of quantitative relations between P–gp specificity and the molecular properties of the substrates. The apparent affinities obtained for the amphiphiles considered in this work suggested a strong increase in affinity with the increase in lipophilicity, while the intrinsic binding affinities are similar. A major concern when using the total ligand concentration is that the apparent affinities obtained depend on the extent of sequestration by the lipid portion of the membrane (due to changes in the amount of membranes, in the lipid composition, and/or in the lipid-to-protein ratio). This was observed in this work through the different apparent affinities obtained from the ATPase activity and the IAAP displacement assays, the later with a larger amount of native membranes. Because the formalism proposed explicitly considers the sequestration of the ligand by the membrane, similar values are obtained for the intrinsic affinities of the amphiphiles from both assays.

It is also shown that the effects on ATPase activity and IAAP displacement can only be adequately described when it is considered that P–gp may accommodate several molecules of the ligand in its binding pocket. A good estimate of this maximum occupancy number is obtained through the comparison between the partition coefficient from the aqueous media to P–gp binding pocket (calculated by closing the thermodynamic circle for interaction with P–gp mediated by the membrane, Figure 9) and that obtained from the macroscopic binding constant (Equation (3)).

The analysis of the ATPase activity and IAAP displacement assays with the formalism proposed in this work shows that the intrinsic affinity for P–gp is similar for all NBD amphiphiles studied in spite of their very different lipophilicity. This shows that bulky hydrophobic groups interact more favorably with the lipid portion of the membrane than with P–gp’s binding pocket.

Finally, a mechanism is proposed for the inhibition of ATPase activity by molecules with bulky non-polar groups and a large amphiphilic moment. This molecular property will lead to a localization at the interface between P–gp’s binding pocket and the lipid bilayer. The binding of several molecules to P–gp leads to the formation of a ring that anchors P–gp to the surrounding lipid bilayer, inhibiting the conformational dynamics required for the P–gp catalytic cycle of ATP hydrolysis and substrate transport.

To further validate (or challenge) the hypothesis proposed in this work, it would be important to characterize the effect on ATPase activity and the structural characterization of P–gp reconstituted in well defined membrane systems for different (but known) local concentrations of P–gp in the membrane, as well as for amphiphiles with distinct overall lipophilicity and amphiphilic moment.

## Figures and Tables

**Figure 1 pharmaceutics-15-00174-f001:**
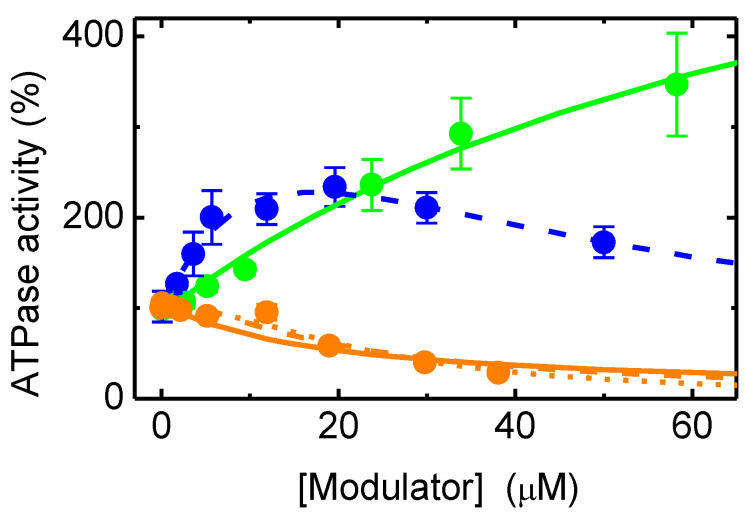
Effect of NBD–C4 (●, **─**), NBD–C8 (●, ▪
▪
▪), and NBD–LysoMPE (●, **─**, ▪
▪
▪, ⋯ ) on P–gp ATPase activity. The symbols represent the average of three to five independent experiments, and the standard deviation is shown as error bars (sometimes smaller than the symbol size). The lines are the best fits of the usual formalism that considers activation and uncompetitive inhibition on P–gp’s ATPase activity depending on the concentration of the modulator (Appendix A). The continuous lines consider only activation (for NBD–C4) or only inhibition (for NBD–LysoMPE), while the dashed lines consider both activation and inhibition. For NBD–LysoMPE, two situations are shown: the best fit imposing $K_{1}\leq K_{2}$ (▪
▪
▪) and no constraint (⋯). See Appendix A for the parameter values. The basal ATPase activity is taken as 100%.

**Figure 2 pharmaceutics-15-00174-f002:**
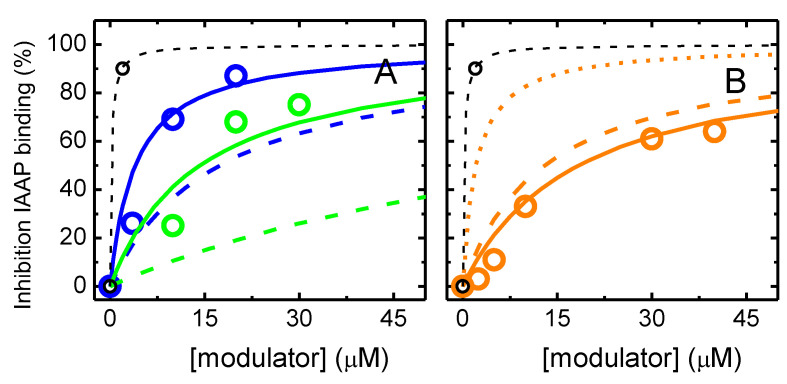
Inhibition of IAAP binding by NBD–C4 (◯) and NBD–C8 (◯), plot **A**, and by NBD–LysoMPE (◯), plot **B**. The results obtained with Tariquidar are also shown (◯). The continuous lines correspond to the best fits of Appendix A, which considers competitive inhibition. The dashed lines are the prediction from the highest affinity obtained in the ATPase assay, with the constraint $K_{1}\leq K_{2}$ (long dash) or no constraint (short dash).

**Figure 3 pharmaceutics-15-00174-f003:**
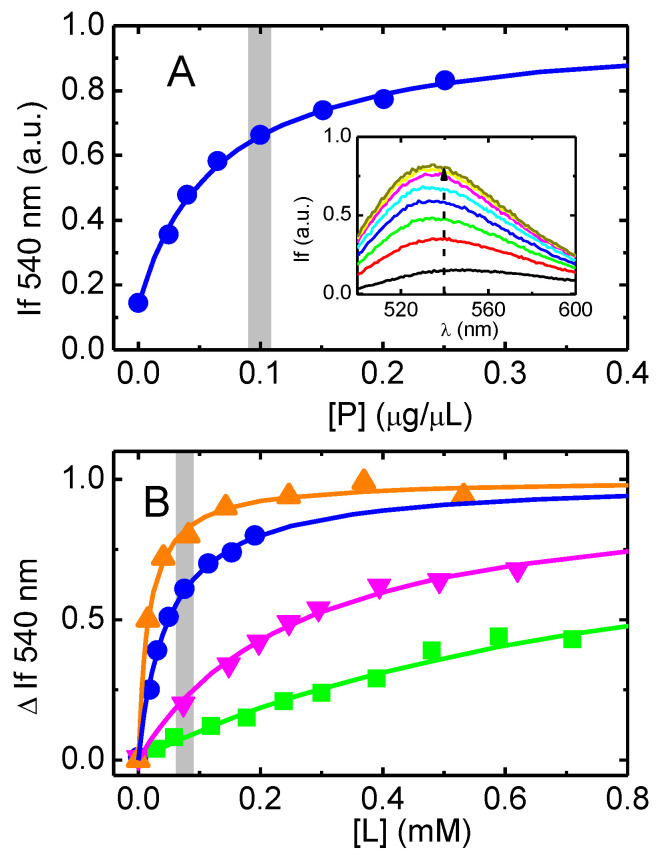
Effect of the native membranes on the NBD amphiphile fluorescence: NBD–C8 (●, **─**), NBD-C6 (▼, **─**), NBD–C4 (■, **─**), and NBD–LysoMPE (▲, **─**). Plot (**A**) shows the variation in the fluorescence spectra (inset) and in the fluorescence intensity at 540 nm for NBD–C8. Panel (**B**) shows data for all amphiphiles studied. The gray bar indicates the concentration of protein in the native membranes used in the ATPase activity assays, and it was 10× higher in the IAAP displacement assays. The lines are the best fits considering simple partition, Appendix A, with the parameters in Table 2.

**Figure 4 pharmaceutics-15-00174-f004:**
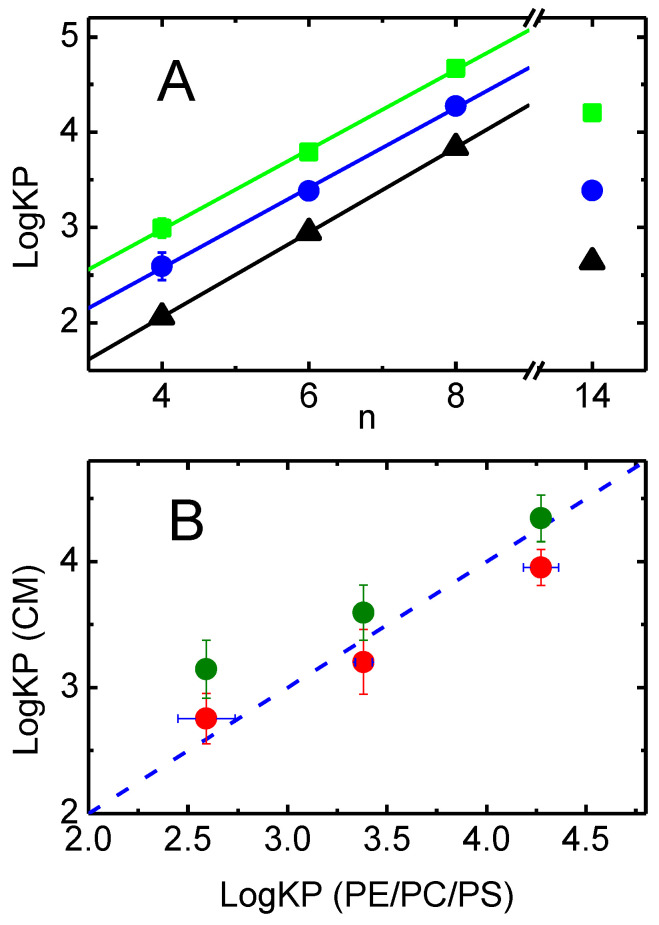
Lipophilicity of the NBD amphiphiles. Plot **A**: Dependence of the calculated LogD (**─**, **▲**) and experimental partition to lipid bilayer composed of POPC (**─**, **■**) or PE:PC:PS 45:35:20 (**─**, **●**), with the length of the alkyl chain. Note the x-axis break between NBD–C8 (*n* = 8) and NBD–LysoMPE (*n* = 14). Plot **B**: Relation between the affinity of the NBD–Cn amphiphiles to the native membranes and to the representative model membranes, considering only the lipids, $V_{M}=V_{L}$ (●) or both the lipids and the proteins $V_{M}=V_{L}+V_{P}$ (●). The identity line is also shown (**- -**). The symbols are the average of at least five independent measurements, and the standard deviations are also shown (some are smaller than the symbol size).

**Figure 5 pharmaceutics-15-00174-f005:**
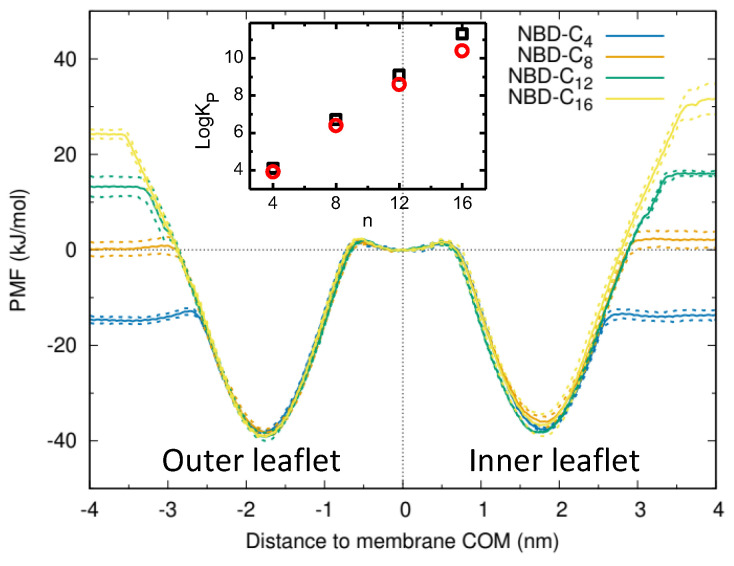
PMF profiles for the interaction of NBD–Cn with the asymmetric lipid membrane (without P–gp). The error estimates are shown by the dotted lines. The inset shows the dependence of the partition coefficient with the length of the amphiphile alkyl chain. See text and Table 1 for details.

**Figure 6 pharmaceutics-15-00174-f006:**
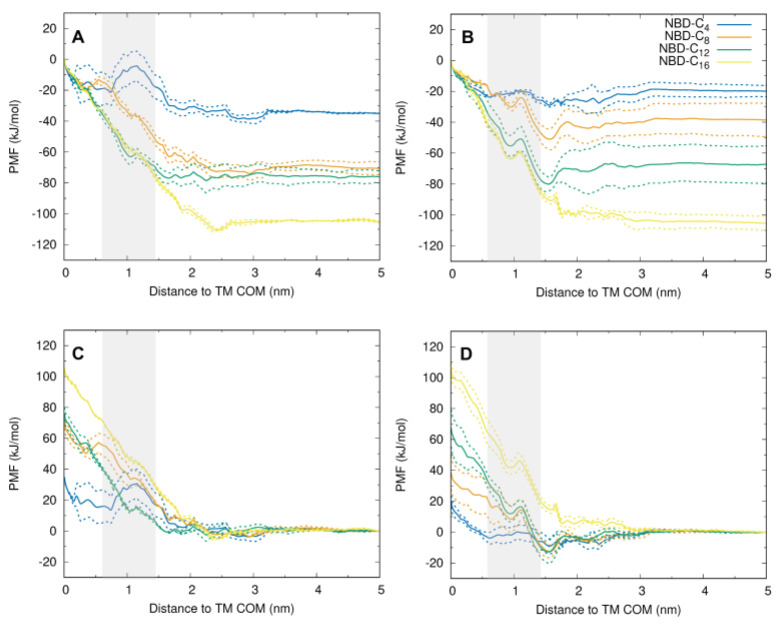
PMF profiles for the transfer of NBD–Cn from the complex asymmetric (**A**,**C**) or from the POPC (**B**,**D**) lipid bilayers to membrane embedded P–gp. In plots (**A**,**B**), the PMF energy reference is defined at 0 nm from the TM COM of P–gp, while in plots (**C**,**D**), the reference is defined in the bulk membrane (5 nm). The gray shading denotes the gate region of P–gp. Error estimates are shown as dotted lines.

**Figure 7 pharmaceutics-15-00174-f007:**
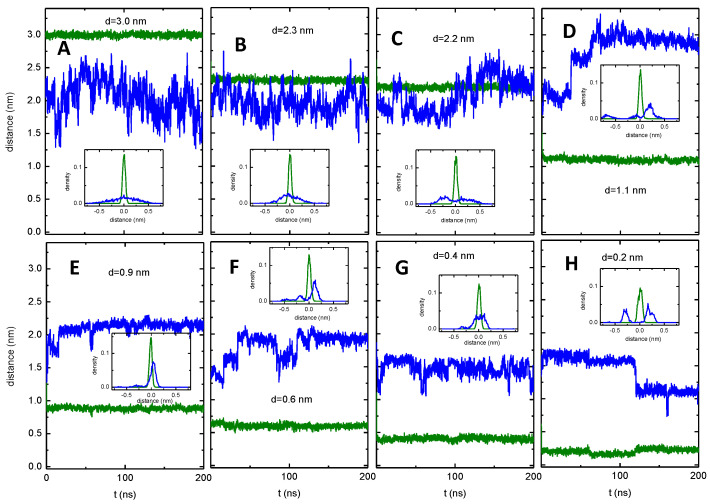
Transverse location of the NBD group of NBD–C4 (*z*, **—**) and radial distance to the TM COM of P–gp (*d*, **—**) as a function of the simulation time in the windows corresponding to a restrained distance to the TM COM of P–gp (*d* = 3 nm (**A**), in the lipid bilayer; *d* = 2.2 and 2.3 nm (**B**,**C**), approaching the P–gp outer surface; *d* = 1.1 nm and 0.9 nm (**D**,**E**), near P–gp’s entry gate; and *d* = 0.6, 0.4 and 0.2 nm (**F**,**G**,**H**), inside the P–gp binding pocket). The insets show the density of the NBD group around its average position in each window relative to the transverse (z, **—**) and to the constrained radial (*d*, **—**) distance.

**Figure 8 pharmaceutics-15-00174-f008:**
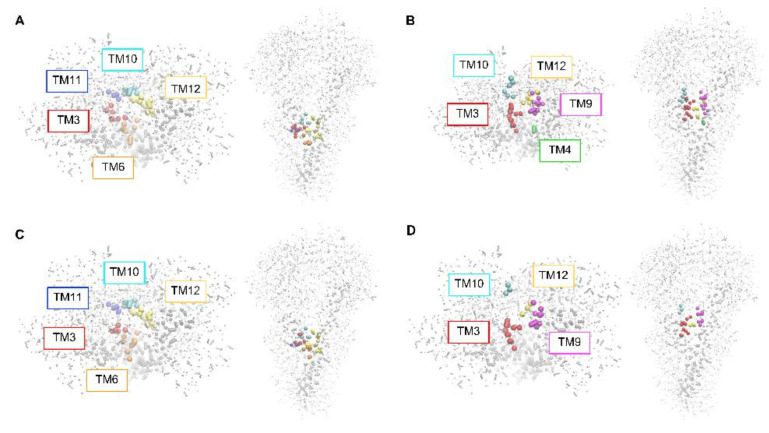
P–gp residues that interacted with the NBD–Cn molecules during the 0.0 nm umbrella window simulation, for NBD–C4 (**A**), NBD–C8 (**B**), NBD-C12, (**C**) and NBD–C16 (**D**). P–gp is shown in gray, with the TM4/6 gate in a darker shade. Residues from TM3 are highlighted in red, from TM4 in green, TM6 in orange, TM9 in pink, TM10 in light blue, TM11 in dark blue, and TM12 in yellow. In each panel, the protein is shown both in the upper view from NB–domains (**left**) and in the front view (**right**).

**Figure 9 pharmaceutics-15-00174-f009:**
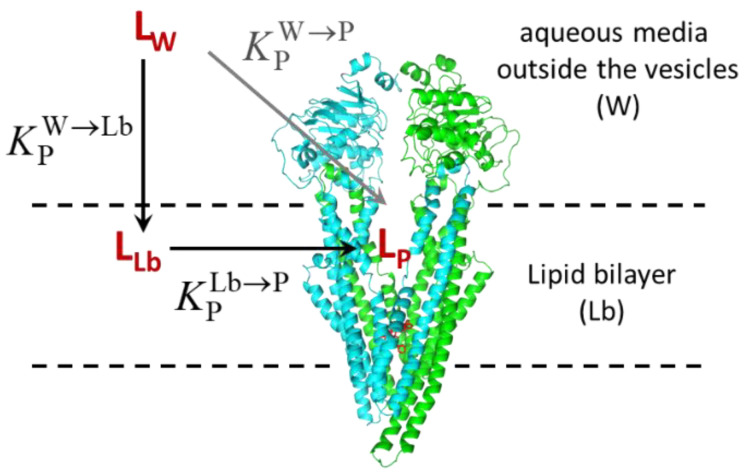
Orientation of P–gp in the inside-out vesicles of native membranes obtained from the High-Five insect cells, with the definition of the distinct environments: aqueous media (W), lipid bilayer (Lb), and P–gp (P). The partition coefficients between the distinct media are also indicated.

**Figure 10 pharmaceutics-15-00174-f010:**
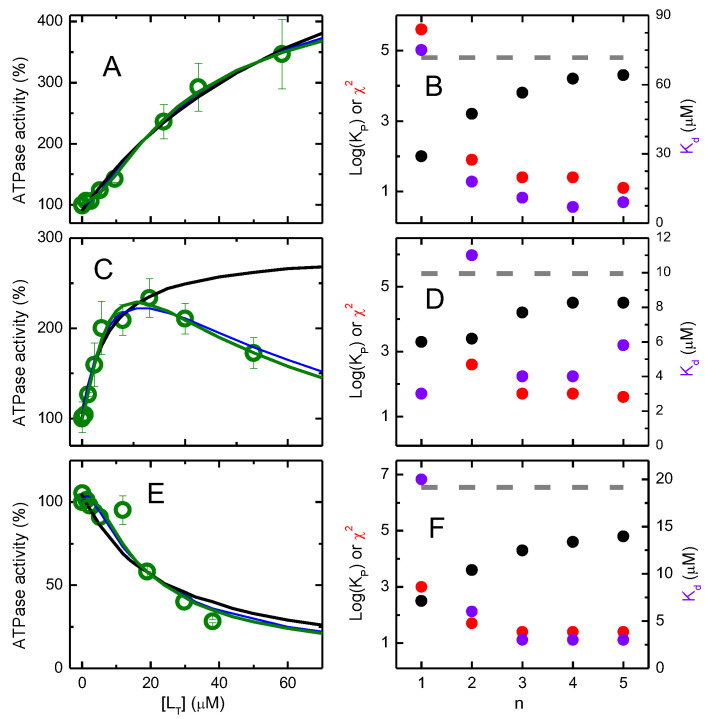
Effect of NBD–C4 (plot **A**,**B**), NBD–C8 (plot **C**,**D**), and NBD–LysoMPE (plot **E**,**F**) on P–gp’s ATPase activity. The plots on the left show the experimental results (**◯**) and the best fit of Equation (9) considering different number of binding sites in P–gp’s binding pocket: n = 1 (**─**), n = 2 (**─**), and n = 5 (**─**). The plots on the right show the effect of the number of binding sites on the quality of the best fit (χ^2^, ●), on the value of the microscopic dissociation constant ($K_{d}$, ●), and on the partition coefficient calculated from *β*_1_ (●). The partition coefficient between the aqueous phase and P–gp obtained from the partition experiments is also shown (▪
▪
▪).

**Figure 11 pharmaceutics-15-00174-f011:**
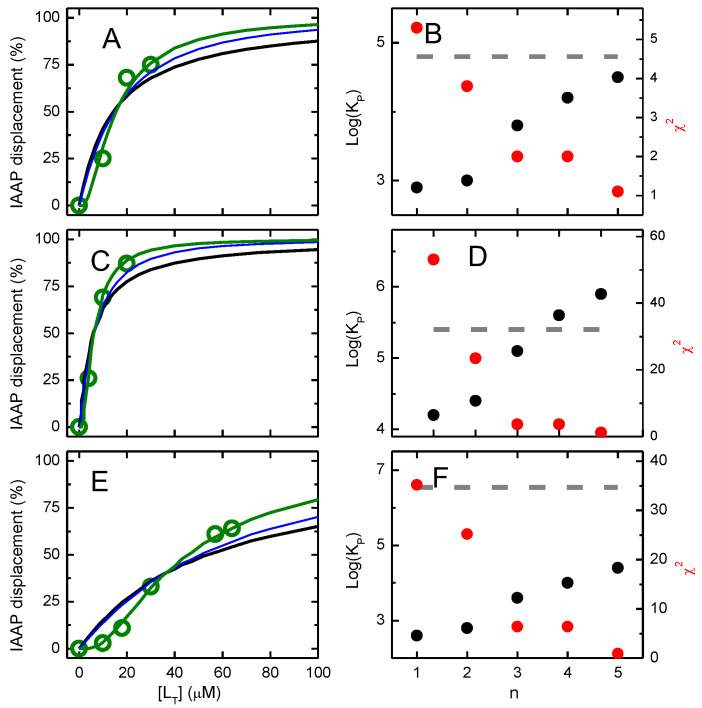
Effect of NBD–C4 (plot (**A**,**B**)), NBD–C8 (plot (**C**,**D**)), and NBD–LysoMPE (plot (**E**,**F**)) on IAAP displacement from P–gp’s binding pocket considering that IAAP is displaced from P–gp’s binding pocket when the number of NBD amphiphiles bound (i) is ≥ n−1. The plots on the left show the experimental results (**◯**) and the best fit of Equation (11) considering different number of binding sites in P–gp’s binding pocket: n = 1 (**─**), n = 2 (**─**), and n = 5 (**─**). The plots on the right show the effect of the number of binding sites on the quality of the best fit (χ^2^, ●) and on the partition coefficient calculated from *β*_1_ (●). The partition coefficient between the aqueous phase and P–gp obtained from the partition experiments is also shown (▪
▪
▪).

**Figure 12 pharmaceutics-15-00174-f012:**
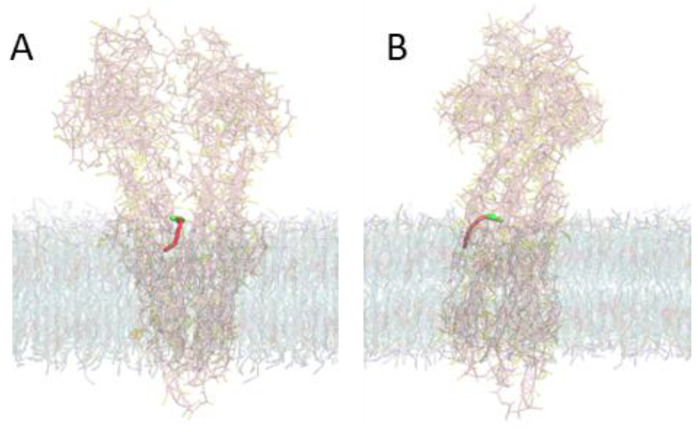
Snapshot of NBD–C16 with the NBD COM restricted at *d* = 0.5 nm from P–gp’s TM COM, obtained at the end of the 200 ns MD simulation. In plot (**A**), the P–gp’s gate between TM4/6 is facing to front, and, in plot (**B**), it is facing to the left side.

**Table 1 pharmaceutics-15-00174-t001:** Name, structure, and common molecular descriptors **^1^** for the P–gp ligands studied in this work.

Name	Structure	CLogP	CLogD_7.4_	Z_7.4_
NBD–C4	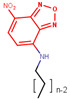	2.06	0
NBD-C6	2.95
NBD–C8	3.84
NBD–LysoMPE	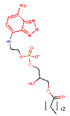	5.01	2.64	−1
IAAP	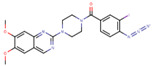	4.17	4.05	0 (97%)+1 (3%)

^1^ Calculated using the MarvinSketch software (version 22.9.0, http://www.chemaxon.com, accessed on 26 October 2022).

## Data Availability

Original data will be provided when requested.

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
