# Peer review of "Interaction of a Homologous Series of Amphiphiles with P-glycoprotein in a Membrane Environment—Contributions of Polar and Non-Polar Interactions"

_pharmaceutics, 2023, doi:10.3390/pharmaceutics15010174_

Round 1
Reviewer 1 Report
Review, Moreno et al.
The aim of the authors was to explore the role of polar and non‐polar interactions of NBD (nitrobenzoxadiazole) - labelled amphiphiles with the efflux transporter P‐glycoprotein (P‐gp). The interaction of these amphiphiles with appropriate model membranes and the whole native membrane, containing the transporter, were characterized by fluorescence measurements. The interaction with P‐glycoprotein was evaluated through effects on ATPase activity and efficiency in inhibition of [125I]‐IAAP binding. The results were complemented with MD simulations to obtain details on the interactions established with the lipid bilayer and P‐gp.
The authors conclude that an increase in the lipophilicity and amphiphilicity lead to a more efficient association with the lipid bilayer, which maintains the non‐polar groups of the amphiphiles, while the polar groups interact with P‐gp’s binding pocket. They also propose a mechanism for inhibition of P‐pg due to the presence of several amphiphiles in this orientation.
In contrast to the clarity of the Abstract, the Results and Discussion sections are far less clear. The very limited number of data points and the inconsistent evaluation procedures do not support the conclusions made in the Abstract.
To simplify the discussion, we collected the main data dispersed throughout the ms in a Table.
|
Compound |
From Tab. 2 |
From. Tab. 2 |
From Tab. 1 |
From Tab. 2 |
From Text |
From Text |
|
K (W to Lb) |
Log(K(W to Lb) |
LogP (calcu.) |
K(Lb to P) |
K1 |
CAC |
|
|
(M) |
(M) |
|||||
|
NBD-C4 |
3.90E+02 |
2.59 |
2.69 |
176 |
8.40E-05 |
2.50E-04 |
|
NBD-C6 |
2.40E+03 |
3.38 |
3.58 |
--------- |
---------- |
---------- |
|
NBD-C8 |
1.90E+04 |
4.28 |
4.46 |
12 |
1.70E-05 |
1.00E-06 |
Major problems:
(1) Membrane binding was studied for three compounds. The interaction with P-gp was studied only for two compounds of the homologous series (NBD-C4 and NBD-C8). The forth compound (NBD-LysoMPE) has a negative charge and cannot be directly compared.
(2) Equation 1: Binding constants are generaly multiplied not added.
Equation 2 (quotient) is more reasonable.
(3) Binding constant of the polar group to the transporter K(Lb to P). As the polar head groups are identical, similar values would be expected for K(Lb to P), however.which is not the case. A lengthy and confusing explanation for the difference between the two values is given by the authors (Lines 764 – 773):
“It is observed that the relative affinity of the amphiphiles from the aqueous medium towards the proteins in the membrane K (W-P) increases as the amphiphile lipophilicity increases (from n=4 to 8). However, this does not accompany the increase observed in their affinity for the lipid bilayer. Therefore, a decrease is observed in the affinity for the membrane protein, relative to that of the lipid bilayer K (Lb P). This is in agreement with the results obtained by MD simulations (Figure 8). At low amphiphile concentrations, away from saturation of the lipid bilayer and proteins, the fraction of amphiphile associated with each medium may be calculated from the respective partition coefficient and volume. It is found that 72% of NBD‐C4 in the membrane is associated with proteins, this decreasing to 15% for NBD‐C8.”
A more likely explanation is that NBD-C8 is partially aggregated because the CAC (critical aggregation concentration) is about 10 fold lower than K1 (concentration of half-maximum P-glycoprotein activation) (see Table A).
(4) The quality of fits to the kinetic two-site binding model used, depends on the quality of data (as always in fitting procedures). In the case of NBD-C8 association problems may again play a role and in the case of the more hydrophilic NBD-C4 inhibition could not be measured.
(5) Fig. 4A and Tab.1: Log P (LogD) data do not agree.
(6) Figure 4B: surprising, that Log P values for lipids only are higher than for lipids + proteins. This would mean that proteins do not substantially bind these amphiphiles. Log P values should be volume independent (corrected for volume).
(7) Lines 663 to 710: Problems discussed qualitatively could be solved by measuring proper biding constants. To give binding in % changes is not helpful.
(8) Although, binding of several (more than two) small molecules seems possible, the model proposed for P-gp inhibition is not consistent with the numerous P-gp structures with inhibitors bound, shown recently in several publications.
(9) The conclusions made by the author (of the second part of the ms) do not correspond to the conclusions expected by the author of the abstract.
This ms is most likely written by two different authors and seems hastily assembled. Neither the experimental data nor the discussion warrants a publication at the current stage.
Author Response
We thank the reviewer for the careful reading of the manuscript. Please see the detailed response in the attached document.

Reviewer 2 Report
The manuscript is very well written and the research work well performed. The authors studied the role of polar and non‐polar interactions of amphiphile molecules with the efflux transporter P‐glycoprotein (P‐gp). This was evaluated using an array of experiments to quantify the effects on ATPase activity, efficiency in inhibition of [125I]‐IAAP binding, and partition to the whole native membranes containing the transporter. Computer-based simulations were also performed to further study the interactions of the molecules with the lipid membrane and pgp protein. This work is very important for early drug development to help predict the potential of a compound to interact with the pgp tranposrter .
Author Response
We thank the reviewer for reading the manuscript and for the positive and encouraging words about our work.
Reviewer 3 Report
The study by Maria J. Moreno et al. investigates the interaction of a series of fluorescent amphiphilic molecules with the efflux transporter P‐glycoprotein in order to clarify the molecular determinants of substrate specificity. These coumpounds have a common polar moiety but a variable alkyl chain length. Despite discripancies at first glance between the effects of these amphiphiles on ATPase activities and IAAP competitive binding, the authors made great efforts to deeply analyze their results and reconcile their experimental observations. What was key in this work was to determine the respective amount of amphiphiles in aqueous media, lipid bilayer, and bound to P‐gp in the different experimental conditions. Although such idea and the phenomenon of drug concentration in the lipid bilayer are not novel since they were notably described by the group of Seelig, the authors took advantage of the fluorescent properties of the molecules to quantify, as accurately as possible, the partition parameters. An other important aspect of the work was to improve the quality of the fits by simulating the most appropriate number of amphiphilic molecules bound to the P-gp. This work shows that the intrinsic affinity for P‐gp is similar for all NBD amphiphiles studied, in spite of their very different lipophilicity. This shows that bulky hydrophobic groups do not interact favorably with P‐gp’s binding pocket, preferably remaining in contact with the lipid bilayer. Second, the authors proposed a mechanism for the inhibition of ATPase activity by molecules with bulky non‐polar groups. This molecular property may lead to a localization at the interface between P‐gp’s binding pocket and the lipid bilayer. They suggest that the binding of several molecules to P‐gp leads to the formation of a ring that anchors P‐gp to the surrounding lipid bilayer, inhibiting the conformational dynamics required for the P‐gp catalytic and transport cycle.
Overall, the biochemical data that have been carefully and in-depth analysed, but i have to stress that i do not have the expertise to judge the molecular dynamic content of the manuscript. The manuscript is well documented and put into context of the existing literature. In my opinion, the work has been seriously conducted and the results certainly complement and expand our knowledge on the molecular determinants of substrate specificity for P-glycoprotein. I only have minor comments to be adressed:
- Lines 311-315 « P‐gp activation (K1) being 84 μM for NBD‐C4 and 17 μM for NBD‐C8 ». From Fig. 1, 17 uM is not near half stimulation ? Similarly, « maximum activation (V1) obtained was high for NBD‐C4 (734 % of the basal ATPase activity), and intermediate for NBD‐C8 (494 %) » does not appears to be around 494% for NBD-C8, but rather near 250% ?
- An important publication from Anna Seelig’s group should be cited at line 647 (Gatlik-Landwojtowicz et al., Biochemistry 2006)
- Although the proposed mechanism for the inhibition of ATPase activity is here plausible, I’m afraid it relies mostly on molecular dynamic simulations. Could you envision other possible mechanisms ? Indeed there might be several ways for the amphiphiles to inhibit the conformational dynamics of P-gp, such as binding and stabilizing the outward-facing state of the protein that occurs during the basal ATPase activity of the transporter. Could you discuss that ?
- Although i leave it to the authors to decide, a big part of the discussion contains experimental analysis (while referring to Fig. 4 to 12 !), I think it would be better to transfer these sections to the results…
Author Response
We thank the reviewer for reading the manuscript and for the positive and encouraging words about our work. Please see the detailed response in the attached document.
